# FedL2P: Federated Learning to Personalize

**Royson Lee[1,2]\*, Minyoung Kim[2], Da Li[2]**
**Xinchi Qiu[1], Timothy Hospedales[2,3], Ferenc Huszár[1], Nicholas D. Lane[1,4]**

[1] University of Cambridge, UK     [2] Samsung AI Center, Cambridge, UK
[3] University of Edinburgh, UK     [4] Flower Labs

## Abstract

Federated learning (FL) research has made progress in developing algorithms for distributed learning of global models, as well as algorithms for local personalization of those common models to the specifics of each client's local data distribution. However, different FL problems may require different personalization strategies, and it may not even be possible to define an effective one-size-fits-all personalization strategy for all clients: depending on how similar each client's optimal predictor is to that of the global model, different personalization strategies may be preferred. In this paper, we consider the federated meta-learning problem of learning personalization strategies. Specifically, we consider meta-nets that induce the batch-norm and learning rate parameters for each client given local data statistics. By learning these meta-nets through FL, we allow the whole FL network to collaborate in learning a customized personalization strategy for each client. Empirical results show that this framework improves on a range of standard hand-crafted personalization baselines in both label and feature shift situations.[0]

## 1 Introduction

Federated learning (FL) is an emerging approach to enable privacy-preserving collaborative learning among clients who hold their own data. A major challenge of FL is to learn from differing degrees of statistical data heterogeneity among clients. This makes it hard to reliably learn a global model and also that the global model may perform sub-optimally for each local client. These two issues are often dealt with respectively by developing robust algorithms for learning the global model [33, 32, 39] and then offering each client the opportunity to personalize the global model to its own unique local statistics via fine-tuning. In this paper, we focus on improving the fine-tuning process by learning a personalization strategy for each client.

A variety of approaches have been proposed for client personalization. Some algorithms directly learn the personalized models [62, 43], but the majority obtain the personalized model after global model learning by fine-tuning techniques such as basic fine-tuning [46, 48], regularised fine-tuning [34, 59], and selective parameter fine-tuning [17, 11, 2, 37]. Recent benchmarks [45, 9] showed that different personalized FL methods suffer from lack of comparable evaluation setups. In particular, dataset- and experiment-specific personalization strategies are often required to achieve state-of-the-art performance. Intuitively, different datasets and FL scenarios require different personalization strategies. For example, scenarios with greater or lesser heterogeneity among clients, would imply different strengths of personalization are optimal. Furthermore, exactly how that personalization should be conducted might depend on whether the heterogeneity is primarily in marginal label shift, marginal feature shift, or conditional shift. None of these facets can be well addressed by a one size fits all personalization algorithm. Furthermore, we identify a previously understudied issue: even

---

\*Corresponding Author: `dsrl2@cam.ac.uk`

[0]Code is available at https://github.com/royson/fedl2p

for a single federated learning scenario, heterogeneous clients may require different personalization strategies. For example, the optimal personalization strategy will have client-wise dependence on whether that client is more or less similar to the global model in either marginal or conditional data distributions. Existing works that learn personalized strategies through the use of personalized weights are not scalable to larger setups and models [55, 14]. On the other hand, the few studies that attempt to optimize hyperparameters for fine-tuning do not sufficiently address this issue as they either learn a single set of personalization hyperparameters [65, 22] and/or learn a hyperparameter distribution without taking account of the client's data distribution [30].

In this paper, we address the issues above by considering the challenge of federated meta-learning of personalization strategies. Rather than manually defining a personalization strategy as is mainstream in FL [27, 9], we use hyper-gradient meta-learning strategies to efficiently estimate personalized hyperparameters. However, apart from standard centralised meta-learning and hyperparameter optimization (HPO) studies which only need to learn a single set of hyperparameters, we learn meta-nets which inductively map from local client statistics to client-specific personalization hyperparameters. More specifically, our approach FedL2P introduces meta-nets to estimate the extent in which to utilize the client-wise BN statistics as opposed to the global model's BN statistics, as well as to infer layer-wise learning rates given each client's metadata.

Our FedL2P thus enables per-dataset/scenario, as well as per-client, personalization strategy learning. By conducting federated meta-learning of the personalization hyperparameter networks, we simultaneously allow each client to benefit from its own personalization strategy, (e.g., learning rapidly, depending on similarity to the global model), and also enable all the clients to collaborate by learning the overall hyperparameter networks that map local meta-data to local personalization strategy. Our FedL2P generalizes many existing frameworks as special cases, such as FedBN [35], which makes a manual choice to normalize features using the client BN's statistics, and various selective fine-tuning approaches [37, 48, 11, 2], which make manual choices on which layers to personalize.

## 2    Related Work

Existing FL works aim to tackle the statistical heterogeneity of learning personalized models by either first learning a global model [46, 39, 32, 48, 29] and then fine-tuning it on local data or directly learning the local models, which can often be further personalized using fine-tuning. Many personalized FL approaches include the use of transfer learning between global [56] and local models [44], model regularization [34], Moreau envelopes [59], and meta-learning [15, 10]. Besides algorithmic changes, many works also proposed model decoupling, in which layers are either shared or personalized [2, 11, 64, 48, 37, 17, 54] or client clustering, which assumes a local model for each cluster [42, 44, 53, 7]. These methods often rely on or adopt a fixed personalization policy for local fine-tuning in order to adapt a global model or further improve personalized performance. Although there exists numerous FL approaches that propose adaptable personalization policies [55, 40, 14], these works are memory intensive and do not scale to larger setups and models. On the other hand, our approach has a low memory footprint (Appendix C) and is directly applicable and complementary to existing FL approaches as it aims to solely improve the fine-tuning process.

Another line of work involves HPO for FL (or FL for HPO). These methods either learn one set of hyperparameters for all clients [22, 30, 65] or random sample from learnt hyperparameter categorical distributions which does not take into account of the client's meta-data [30]. Moreover, some of these methods [22, 65] search for a set of hyperparameters based on the local validation loss given the initial set of weights prior to the federated learning of the model [22, 65], which might be an inaccurate proxy to the final performance. Unlike previous works which directly learn hyperparameters, we deploy FL to learn meta-nets that take in, as inputs, the client meta-data to generate personalized hyperparameters for a given pretrained model. A detailed positioning of our work in comparison with existing literature can be found in Appendix B.

## 3    Proposed Method

### 3.1    Background & Preliminaries

**Centralized FL.** A typical centralized FL setup using FedAvg [46] involves training a global model $\theta_g$ from $C$ clients whose data are kept private. At round $t$, $\theta_g^{t-1}$ is broadcast to a subset of clients selected, $\tilde{C}$, using a fraction ratio $r$. Each selected client, $i \in \tilde{C}$, would then update the model using

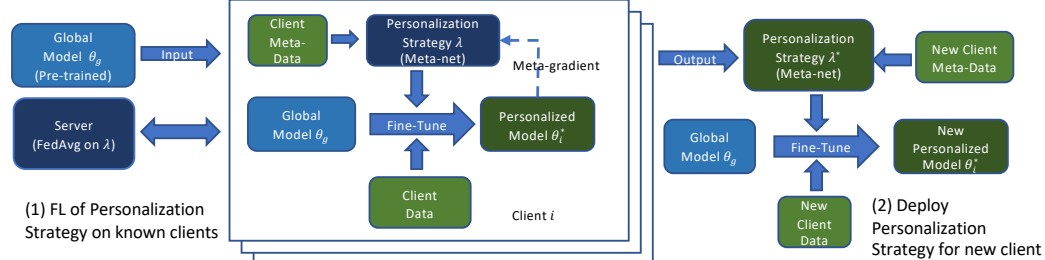

**Figure 1:** FedL2P's workflow. For each client, the meta-net (parameterized by $\lambda$) takes the client meta-data (e.g., local data profile) and outputs hyperparameters. We update $\lambda$ by optimizing the meta objective which is the validation loss of the model finetuned with the hyperparameters returned by the meta-net. The updated personalization strategies $\lambda^*$ from clients are collected/aggregated (via FedAvg) in the server for the next round.

a set of hyperparameters and its own data samples, which are drawn from its local distribution $P_i$ defined over $\mathcal{X} \times \mathcal{Y}$ for a compact space $\mathcal{X}$ and a label space $\mathcal{Y}$ for $e$ epochs. After which, the learned local models, $\theta_i^{t-1}$, are sent back to the server for aggregation $\theta_g^t = \sum_i^{\tilde{C}} \frac{N_i}{\sum_{i'} N_{i'}} \theta_i^{t-1}$ where $N_i$ is the total number of local data samples in client $i$ and the resulting $\theta_g^t$ is used for the next round. The aim of FL is either to minimize the global objective $\mathbb{E}_{(x,y) \sim P} \mathcal{L}(\theta_g; x, y)$ for the global data distribution $P$ or local objective $\mathbb{E}_{(x,y) \sim P_i} \mathcal{L}_i(\theta_i; x, y)$ for all $i \in C$ where $\mathcal{L}_i(\theta; x, y)$ is the loss given the model parameters $\theta$ at data point $(x, y)$. As fine-tuning is the dominant approach to either personalize from a high-performing $\theta_g$ or to optimize $\theta_i$ further for each client, achieving competitive or state-of-the-art results in recent benchmarks [45, 9], we focus on the collaborative learning of meta-nets which generates a personalized set of hyperparameters used during fine-tuning to further improve personalized performance without compromising global performance.

**Non-IID Problem Setup.** Unlike many previous works, our method aims to handle both common label and feature distribution shift across clients. Specifically, given features $x$ and labels $y$, we can rewrite the joint probability $P_i(x, y)$ as $P_i(x|y)P_i(y)$ and $P_i(y|x)P_i(x)$ following [28]. We focus on three data heterogeneity settings found in many realistic settings: both label & distribution skew in which the marginal distributions $P_i(y)$ & $P_i(x)$ may vary across clients, respectively, and concept drift in which the conditional distribution $P_i(x|y)$ may vary across clients.

### 3.2 FedL2P: FL of Personalization Strategies

We now present our proposed method, FedL2P, to tackle collaborative learning of personalization strategies under data heterogeneity. Our main motivation is that the choice of the hyperparameters for the personalized fine-tuning, such as the learning rates and feature mean and standard deviations (SD) statistics of the batch normalization [26] (BN) layers, is very crucial. Although existing FL-HPO approaches aim to learn these hyperparameters *directly*[2] [22, 65, 30], we aim to do it in a meta learning or hypernetwork-like fashion: learning a neural network (dubbed *meta-net*) that takes certain profiles of client data (e.g., summary statistics of personalized data) as input and outputs near-optimal hyperparameters. The meta-net is learned collaboratively in an FL manner without sharing local data. The returned hyperparameters from the meta-net are then deployed in client's personalized fine-tuning. The main advantage of this meta-net approach over the direct HPO is that a brand new client needs not do time-consuming HPO, but just gets their optimal hyperpameters by a single feed-forward pass through the meta-net. Our idea is visualized in Fig. 1. For each FL round, the latest meta-net is distributed to the participating clients; each client then performs meta learning to update the meta-net; the local updated meta-nets are sent to the server for aggregation. Details of our algorithm are described below.

**Hyperparameter Selection.** There is a wide range of local training hyperparameters that can be optimized, some of which were explored in previous FL works [22, 65, 30]. As local training is often a costly process, we narrowed it down to two main sets of hyperparameters based on previous works that showed promising results in dealing with non-IID data: *BN hyperparameters* and *selective update hyperparameters*.

**Batch Normalization Hyperparameters.** The first set of hyperparameters involves BN and explicitly deals with feature shift. Notably, FedBN [35] proposed keeping the BN layers local and the other layers global to better handle feature shift across clients; BN has found success at mitigating domain

---

[2]To our knowledge, no previous FL-HPO works learn BN statistics, crucial for dealing with feature shifts.

shifts in various domain adaptation tasks [36, 8] and can be formulated as follows:

$$g(x) = \frac{x - \hat{\mu}}{\sqrt{\hat{\sigma}^2 + \epsilon}} * \gamma + \delta \tag{1}$$

where $g$ is a BN layer, $x$ is its input features, $(\hat{\mu}, \hat{\sigma}^2)$ are the estimated running mean, variance, and $\epsilon$ is used for numerical stability. During training, the batch statistics $E(x)$ and $Var(x)$ are used instead, keeping running estimates of them, $\hat{\mu}$ and $\hat{\sigma}^2$. During inference, these estimates are used[3]. Both $\gamma$ and $\delta$ are learnable parameters used to scale and shift the normalized features.

Although deploying local BN layers is useful to counteract the drawbacks of feature shift, using global BN layers is often beneficial for local fine-tuning in cases where the feature shift is minimal as it helps speed-up convergence and reduces overfitting. Hence, we propose learning a hyperparameter $\beta$ for each BN layer as follows:

$$\hat{\mu} = (1 - \beta) * \hat{\mu}_{pt} + \beta * \hat{\mu}_i, \qquad \hat{\sigma}^2 = (1 - \beta) * \hat{\sigma}_{pt}^2 + \beta * \hat{\sigma}_i^2 \tag{2}$$

where $\hat{\mu}_{pt}$ and $\hat{\sigma}_{pt}^2$ are the estimated running mean and variance by the given pretrained model; $\hat{\mu}_i$ and $\hat{\sigma}_i^2$ are the running mean and variance estimated by the client. When $\beta \to 0$, the client solely uses pretrained model's BN statistics and when $\beta \to 1$, it uses its local statistics to normalize features. Thus $\beta$ indicates the degree in which the client should utilize its own BN statistics against pretrained model's, to handle the feature shift in its local data distribution.

**Selective Update Hyperparameters.** A variety of recent personalized FL works achieved promising results by manually selecting and fine-tuning a sub-module of $\theta_g$ during personalization [48, 2, 37], (*e.g.*, the feature extractor or the classifier layers), while leaving the remaining modules in $\theta_g$ frozen. It is beneficial because it allows the designer to manage over- vs under-fitting in personalization. *e.g.*, if the per-client dataset is small, then fine-tuning many parameters can easily lead to overfitting, and thus better freezing some layers during personalization. Alternatively, if the clients differ significantly from the global model and/or if the per-client dataset is larger, then more layers could be beneficially personalised without underfitting. Clearly the optimal configuration for allowing model updates depends on the specific scenario and the specific client. Furthermore, it may be beneficial to consider a more flexible range of hyperparameters that control a continuous degree of fine-tuning strength, rather than a binary frozen/updated decision per module.

To automate the search for good personalization strategies covering a range of wider and more challenging non-IID setups, we consider layer-wise learning rates, $\eta$ for all learnable weights and biases. This parameterization of personalization encompasses all the previous manual frozen/updated split approaches as special cases. Furthermore, while these approaches have primarily considered heterogeneity in the marginal label distribution, we also aim to cover feature distribution shift between clients. Thus we also include the learn-

---

**Algorithm 1** FedL2P: FL of meta-nets for Personalization Hyperparameters

**Input:** Pretrained global model $\theta_g$ with $M$ layers and $B$ BN layers, each has $J$ input channels. Fraction ratio $r$. Total no. of clients $C$. No. of update iterations $K$. Training loss $\mathcal{L}_T$ and validation loss $\mathcal{L}_V$. $\zeta$ is the learning rate for $\lambda$.

1: initialize $\lambda = \{\boldsymbol{w}_{bn}, \boldsymbol{w}_{lr}, \tilde{\boldsymbol{\eta}}\}$
2: **for** round $t = 1, \ldots, T$ **do**
3:  $\tilde{C} \leftarrow$ Random sample $Cr$ clients
4:  **for** client $i \in \tilde{C}$ **do**
5:   send $\theta_g, \lambda$ to client
6:   Forward pass of local dataset to compute
     1) $E(x_m), SD(x_m)$ for $1 \le m \le M - 1$
     2) $\mu_{b,j}, \sigma_{b,j}^2$ for $b = 1, \ldots, B$ and $j = 1, \ldots, J$.
7:   Compute $\xi_b$ for $b = 1, \ldots, B$ using Eq. 3
8:   **for** iteration $k = 1, \ldots, K$ **do**
9:    $\theta_i \leftarrow$ Finetune $\theta_g$ using $\mathcal{L}_T$ for $e$ epochs
10:    $\lambda \leftarrow \lambda - \zeta \ Hypergradient(\mathcal{L}_V, \mathcal{L}_T, \lambda, \theta_i)$
11:   **end for**
12:   send $\lambda$ and num of data samples $N$ to server
13:  **end for**
14:  $\lambda \leftarrow \sum_i^{\tilde{C}} \frac{N_i}{\sum N_i} \lambda_i$
15: **end for**

**Output:** $\lambda$

---

**Algorithm 2** Hypergradient

**Input:** Validation Loss $\mathcal{L}_V$ and training Loss $\mathcal{L}_T$. Learning rate $\psi$ and no. of iterations $Q$. Fixed point $(\lambda', \theta^*(\lambda'))$.

1: $p = v = \partial_\theta \mathcal{L}_V|_{(\lambda', \theta^*(\lambda'))}$
2: **for** iteration $1, \ldots, Q$ **do**
3:  $v \leftarrow v - \psi v \partial_{\theta^*}^2 \mathcal{L}_T$
4:  $p \leftarrow p + v$
5: **end for**

**Output:** $\partial_\lambda \mathcal{L}_V|_{(\lambda', \theta^*(\lambda'))} - p \partial_{\lambda\theta} \mathcal{L}_T|_{(\lambda', \theta^*(\lambda'))}$

---

ing rates for the BN parameters, $\gamma$ and $\delta$, allowing us to further tackle feature shift by adjusting the means and SD of the normalized features.

---

[3]When the running mean & variance are not tracked, the batch statistics is used in both training and inference.

**Hyperparameter Inference for Personalization.** We aim to estimate a set of local hyperparameters that can best personalize the pretrained model for each client given a group of clients whose data might not be used for pretraining. To accomplish this, we learn to estimate hyperparameters based on the degree of data heterogeneity of the client's local data with respect to the data that the model is pretrained on. There are many ways to quantify data heterogeneity, such as utilizing the earth mover's distance between the client's data distribution and the population distribution for label distribution skew [63] or taking the difference in local covariance among clients for feature shift [35]. In our case, we aim to distinguish both label and feature data heterogeneity across clients. To this end, we utilize the client's local input features to each layer with respect to the given pretrained model. Given a pretrained model with $M$ layers and $B$ BN layers, we learn $\boldsymbol{\eta} = \eta_1, \ldots, \eta_{2M}$[4] and $\boldsymbol{\beta} = \beta_1, \ldots, \beta_B$, using two functions, each of which is parameterized as a multilayer perceptron (MLP), named meta-net, with one hidden layer due to its ability to theoretically approximate almost any continuous function [12, 57]. We named the meta-net that estimates $\boldsymbol{\beta}$ and the meta-net that estimates $\boldsymbol{\eta}$ as BNNet and LRNet respectively. Details about the architecture can be found in Appendix. E.

To estimate $\boldsymbol{\beta}$, we first perform a forward pass of the local dataset on the given pretrained model, computing the mean and SD of each channel of each input feature for each BN layer. We then measure the distance between the local feature distributions and the pretrained model's running estimated feature distributions of the $b$-th BN layer as follows:

$$\xi_{i,b} = \frac{1}{J} \sum_{j=1}^{J} \frac{1}{2} \Big( D_{KL}(P_j || Q_j) + D_{KL}(Q_j || P_j) \Big), \tag{3}$$

where $P_j = \mathcal{N}(\mu_{i,b,j}, \sigma_{i,b,j}^2)$, $Q_j = \mathcal{N}(\hat{\mu}_{pt,b,j}, \hat{\sigma}_{pt,b,j}^2)$, $D_{KL}$ is the KL divergence and $J$ is the number of channels of the input feature. $\xi$ is then used as an input to BNNet, which learns to estimate $\boldsymbol{\beta}$ as shown in Eq. 4.

Similarly, we compute the mean and SD of each input feature per layer by performing a forward pass of the local dataset on the pretrained model and use it as an input to LRNet. Following best practices from previous non-FL hyperparameter optimization works [49, 3], we use a learnable post-multiplier $\tilde{\boldsymbol{\eta}} = \tilde{\eta}_1, \ldots, \tilde{\eta}_{2M}$ to avoid limiting the range of the resulting learning rates (Eq 4).

$$\begin{aligned} \boldsymbol{\beta} &= \text{BNNet}(\boldsymbol{w}_{bn}; \xi_1, \xi_2, \ldots, \xi_{B-1}, \xi_B) \\ \boldsymbol{\eta} &= \text{LRNet}(\boldsymbol{w}_{lr}; E(x_0), SD(x_0), E(x_1), SD(x_1) \ldots, E(x_{M-1}), SD(x_{M-1})) \odot \tilde{\boldsymbol{\eta}} \end{aligned} \tag{4}$$

where $\odot$ is the Hadamard product, $x_{m-1}$ is the input feature to the $m$-th layer, and $\boldsymbol{w}_{bn}$ and $\boldsymbol{w}_{lr}$ are the parameters of BNNet and LRNet respectively. $\boldsymbol{\beta}$ is used to compute the running mean and variance in the forward pass for each BN layer as shown in Eq. 2 and $\boldsymbol{\eta}$ is used as the learning rate for each weight and bias in the backward pass. We do not restrict $\tilde{\boldsymbol{\eta}}$ to be positive as the optimal learning rate might be negative [5].

**Federated Hyperparameter Learning.** We deploy FedAvg [46] to federatedly learn a set of client-specific personalization strategies. Specifically, we learn the common meta-net $\lambda = \{\boldsymbol{w}_{bn}, \boldsymbol{w}_{lr}, \tilde{\boldsymbol{\eta}}\}$ that generates client-wise personalization hyperparameters $\{\boldsymbol{\beta}_i, \boldsymbol{\eta}_i\}$, such that a group of clients can better adapt a pre-trained model $\theta_g$ by fine-tuning to their local data distribution. So we solve:

$$\min_\lambda \mathcal{F}(\lambda, \theta_g) = \sum_{i=1}^{C} \frac{N_i}{\sum_{i'} N_{i'}} \mathcal{L}_{i,V}(\theta_i^*(\lambda), \lambda)$$
$$\text{s.t. } \theta_i^*(\lambda) = \arg\min_{\theta_i} \mathcal{L}_{i,T}(\theta_i, \lambda) \tag{5}$$

where $\theta_i^*$ is the set of optimal personalized model parameters after fine-tuning $\theta_g$ for $e$ epochs on the local dataset, $\mathcal{L}_{i,V}(\theta, \lambda) = \mathbb{E}_{(x,y) \sim V_i} \mathcal{L}_i(\theta, \lambda; x, y)$ and $V_i$ is the validation set (samples from $P_i$) for the client $i$ - similarity for $L_{i,T}$.

For each client $i$, the validation loss gradient with respect to $\lambda$, known as the hypergradient, can be computed as follows:

$$d_\lambda \mathcal{L}_V(\theta^*(\lambda), \lambda) = \partial_\lambda \mathcal{L}_V(\theta^*(\lambda), \lambda) + \partial_{\theta^*(\lambda)} \mathcal{L}_V(\theta^*(\lambda), \lambda) \, \partial_\lambda \theta^*(\lambda) \tag{6}$$

To compute $\partial_\lambda \theta^*$ in Eq. 6, we use the implicit function theorem (IFT):

---

[4]We assume all layers have weights and biases here.

$$\partial_\lambda \theta^*|_{\lambda'} = -(\partial_\theta^2 \mathcal{L}_T(\theta, \lambda))^{-1} \partial_{\lambda\theta} \mathcal{L}_T(\theta, \lambda)|_{\lambda', \theta^*(\lambda')} \tag{7}$$

The full derivation is shown in Appendix A. We use Neumann approximation and efficient vector-Jacobian product as proposed by Lorraine et al. [38] to approximate the Hessian inverse in Eq. 7 and compute the hypergradient, which is further summarized in Algorithm 2. In practice, $\theta^*$ is approximated by fine-tuning $\theta_g$ on $\mathcal{L}_T$ using the client's dataset. Note that unlike in many previous works [38, 47] where $\partial_\lambda \mathcal{L}_V$ is often 0 as the hyperparameters often do not directly affect the validation loss, in our case $\partial_{\boldsymbol{w}_{bn}} \mathcal{L}_V \neq 0$.

Algorithm 1 summarizes FedL2P. Given a pretrained model and a new group of clients to personalize, we first initialize $\lambda$ (line 1). For every FL round, we sample $Cr$ clients and send both the parameters of the pretrained model and $\lambda$ to each client (lines 3-5). Each client then performs a forward pass of their local dataset to compute the mean (E) and standard deviation (SD) of the input features to each layer and the statistical distance between the local feature distributions and the pretrained model's running estimated feature distributions for each BN layer (lines 6-7). $\lambda$ is then trained for $K$ iterations; each iteration optimizes the pretrained model on $\mathcal{L}_T$ for $e$ epochs, applying $\boldsymbol{\beta}$ and $\boldsymbol{\eta}$ computed using Eq. 4 (lines 8-9) at every forward and backward pass respectively. Each client then computes the hypergradient of $\lambda$ as per Algorithm. 2 and update $\lambda$ at the end of every iteration (line 10). Finally, after $K$ iterations, each client sends back the updated $\lambda$ and its number of data samples, which is used for aggregation using FedAvg [46] (lines 12-14). The resulting $\lambda$ is then used for personalization: each client finetunes the model using its training set and evaluates it using its test set.

### 3.3 Adapting the Losses for IFT

In the IFT, we solve the following problem:

$$\min_\lambda \mathcal{L}_V(\theta^*(\lambda), \lambda) \text{ s.t. } \theta^*(\lambda) = \arg\min_\theta \mathcal{L}_T(\theta, \lambda). \tag{8}$$

For the current $\lambda$, we first find $\theta^*(\lambda)$ in (8) by performing several SGD steps with the training loss $\mathcal{L}_T$. Once $\theta^*(\lambda)$ is obtained, we can compute the hypergradient $d_\lambda \mathcal{L}_V(\theta^*(\lambda), \lambda)$ by the IFT, which is used for updating $\lambda$. As described in (6) and (7), this hypergradient requires $\partial_\lambda \mathcal{L}_T(\theta, \lambda)$, implying that the training loss has to be explicitly dependent on the hyperparameter $\lambda$. As alluded in Lorraine et al. [38], it is usually not straightforward to optimize the learning rate hyperparameter via the IFT, mainly due to the difficulty of expressing the dependency of the training loss on learning rates. To address this issue, we define the training loss as follows:

$$\mathcal{L}_T(\theta, \lambda) = \mathbb{E}_{(x,y) \sim P_T} CE(f_{\theta', \boldsymbol{\beta}(\lambda)}(x), y) \text{ where} \tag{9}$$

$$\theta' = \theta - \boldsymbol{\eta}(\lambda) \nabla_\theta \mathbb{E}_{(x,y) \sim P_T} CE(f_{\theta, \boldsymbol{\beta}(\lambda)}(x), y). \tag{10}$$

Here $f_{\theta, \boldsymbol{\beta}}(x)$ indicates the forward pass with network weights $\theta$ and the batch norm statistics $\boldsymbol{\beta}$, and $CE()$ is the cross-entropy loss. Note that in (10), we can take several (not just one) gradient update steps to obtain $\theta'$. Now, we can see that $\mathcal{L}_T(\theta, \lambda)$ defined as above has explicit dependency on the learning rates $\boldsymbol{\eta}(\lambda)$. Interestingly, the stationary point $\theta^*(\lambda)$ of $\mathcal{L}_T(\theta, \lambda)$ coincides with $\theta'$, that is, $\theta^*(\lambda) = \theta'$, which allows for a single instance of inner-loop iterations as Line 9 in Alg. 1. Finally, the validation loss is defined as:

$$\mathcal{L}_V(\theta, \lambda) = \mathbb{E}_{(x,y) \sim P_V} CE(f_{\theta, \boldsymbol{\beta}(\lambda)}(x), y),$$

showing clear dependency on BNNet parameters through $\boldsymbol{\beta}(\lambda)$ as discussed in the previous section.

## 4 Evaluation

### 4.1 Experimental Setup

Experiments are conducted on image classification tasks of different complexity. We use ResNet-18 [20] for all experiments and SGD for all optimizers. All details of the pretrained models can be found in Appendix. D. Additionally, the batch size is set to 32 and the number of local epochs, $e$, is set to 15 unless stated otherwise. The learning rate ($\zeta$) for $\lambda = \{\boldsymbol{w}_{bn}, \boldsymbol{w}_{lr}, \tilde{\boldsymbol{\eta}}\}$ is set to $\{10^{-3}, 10^{-3}, 10^{-4}\}$, respectively. The hypergradient is clipped by value

Table 1: FedL2P complements existing FL methods by improving on the finetuning process. Experiments on CIFAR10 ($e = 15$).

| $\alpha$ | Approach | +FT (BN C) | +FedL2P |
|---|---|---|---|
| **1000** | FedAvg | 63.04±0.02 | 65.13±0.02 |
| (↓ heterogeneity) | PerFedAvg(HF) | 34.58±0.13 | 47.58±0.01 |
| | FedBABU | 65.00±0.07 | 66.49±0.03 |
| **1.0** | FedAvg | 61.42±0.13 | 65.76±0.31 |
| | PerFedAvg(HF) | 44.85±0.28 | 50.2±1.26 |
| | FedBABU | 68.92±0.11 | 70.71±0.28 |
| **0.5** | FedAvg | 62.34±0.14 | 68.45±0.5 |
| | PerFedAvg(HF) | 52.43±0.16 | 55.05±0.53 |
| | FedBABU | 72.26±0.1 | 72.87±0.42 |
| **0.1** | FedAvg | 79.15±0.07 | 80.28±0.07 |
| (↑ heterogeneity) | PerFedAvg(HF) | 77.31±0.05 | 77.68±0.13 |
| | FedBABU | 79.50±0.08 | 79.58±0.04 |

Table 2: Experiments on CIFAR-10 using pretrained model trained using FedAvg [46]. Both initial & personalized accuracies are learnt and personalized on the same set of clients.

| $\alpha$ | Epochs (e) | Global Accuracy | +FT (BN C) | +FT (BN G) | +FT (BN I) | +L2P | +FedL2P |
|---|---|---|---|---|---|---|---|
| **1000** | 5 | 65.13 | 64.35±0.03 | 62.14±0.13 | 56.58±0.08 | 53.61±0.12 | **64.53±0.06** |
| (↓ heterogeneity) | 15 | 65.13 | 63.04±0.02 | 59.85±0.04 | 55.72±0.03 | 58.41±0.38 | **65.13±0.02** |
| **1.0** | 5 | 60.19 | 59.72±0.18 | 63.45±0.04 | 50.8±0.06 | 55.81±0.03 | **66.05±0.09** |
| | 15 | 60.19 | 61.42±0.13 | 63.23±0.15 | 54.94±0.07 | 61.77±0.25 | **65.76±0.31** |
| **0.5** | 5 | 57.12 | 58.22±0.02 | 67.16±0.11 | 50.33±0.01 | 59.79±0.09 | **68.96±0.09** |
| | 15 | 57.12 | 62.34±0.14 | 67.4±0.06 | 58.12±0.07 | 65.28±0.23 | **68.45±0.5** |
| **0.1** | 5 | 44.86 | 68.04±0.05 | 78.73±0.04 | 61.91±0.06 | 74.26±0.29 | **80.33±0.13** |
| (↑ heterogeneity) | 15 | 44.86 | 79.15±0.07 | 78.97±0.07 | 75.94±0.0 | 78.6±0.21 | **80.28±0.07** |

$[-1, 1]$, $Q = 3$, and $\psi = 0.1$ in Alg. 2. The maximum number of communication rounds is set to $500$, and over the rounds we save the $\lambda$ value that leads to the lowest validation loss, averaged over the participating clients, as the final learned $\lambda$. The fraction ratio $r = 0.1$ unless stated otherwise, sampling $10\%$ of the total number of clients per FL round. We focus on non-IID labels and feature shifts while assuming that each client has an equal number of samples. Finally, to generate heterogeneity in label distributions, we follow the latent Dirichlet allocation (LDA) partition method [24, 61, 52]: $y \sim Dir(\alpha)$ for each client. Hence, the degree of heterogeneity in label distributions is controlled by $\alpha$; as $\alpha$ decreases, the label non-IIDness increases, and vice versa.

### 4.1.1 Datasets

**CIFAR10 [31].** A widely-used image classification dataset, also popular as an FL benchmark. The number of clients $C$ is set to 1000 and $20\%$ of the training data is used for validation.

**CIFAR-10-C [21].** The test split of the CIFAR10 dataset is corrupted with common corruptions. We used 10 corruption types[5] with severity level 3 and split each corruption dataset into $80\%/20\%$ training/test sets. A subset ($20\%$) of the train set is further held out to form a validation set. Each corruption type is partitioned by $Dir(\alpha)$ among 25 clients, hence $C = 250$.

**Office-Caltech-10 [19] & DomainNet [51].** These datasets are specifically designed to contain several domains that exhibit different feature shifts. We set the no. of samples of each domain to be the smallest of the domains, random sampling the larger domains. For Caltech-10, we set $r = 1.0$ and $C = 4$, one for each domain and thus partitioned labels are IID. For DomainNet, we set $C = 150$, of which each of the 6 domain datasets are partitioned by $Dir(\alpha)$ among 25 clients, resulting in a challenging setup with both feature & label shifts. Following FedBN [35], we take the top 10 most commonly used classes in DomainNet.

**Speech Commands V2 [60].** We use the 12-class version that is naturally partitioned by speaker, with one client per speaker. It is naturally imbalanced in skew and frequency, with 2112, 256, 250 clients/speakers for train, validation, and test. We sampled 250 of 2112 training clients with the most data for our *seen* pool of clients and sampled 50 out of 256+250 validation and test clients for our *unseen* client pool. Each client's data is then split $80\%/20\%$ for train and test sets, with a further $20\%$ of the resulting train set held out to form a validation set.

### 4.1.2 Baselines

We run all experiments three times and report the mean and SD of the test accuracy. We consider three different setups of fine-tuning in our experiments as below. As basic fine-tuning (FT) is used ubiquitously in FL, our experiments directly compare FT with FedL2P in different non-IID scenarios. We are agnostic to the FL method used to obtain the global model or to train the meta-nets.

**FT (BN C).** Client BN Statistics. Equivalent to setting $\beta = 1$ in Eq. 2, thus using client statistics to normalize features during fine-tuning. This is similar to FedBN [35], and is adopted by FL works such as Ditto [34] and PerFedAvg [15].
**FT (BN G).** Equivalent to setting $\beta = 0$ in Eq. 2. BN layers use the pretrained global model's BN statistics to normalize its features during fine-tuning.
**FT (BN I).** BN layers use the incoming feature batch statistics to normalize its features during fine-tuning. This setting is adopted by FL works such as FedBABU [48].
**L2P.** Our proposed method without FL; $\lambda$ is learnt locally given the same compute budget before being used for personalization. Hence, L2P does client-wise HPO independently.

## 4.2 Experiments on Marginal Label Shift

We evaluate our approach based on the conventional setup of personalized FL approaches [9, 45], where a global model, $\theta_g$, is first learned federatedly using existing algorithms and then personalized

---

[5] *brightness, frost, jpeg_compression, contrast, snow, motion_blur, pixelate, speckle_noise, fog, saturate*

Table 3: Personalized test accuracies of CIFAR-10-C, Office-Caltech-10, Domainnet ($e = 15$).

| $\alpha$ | Dataset | +FT (BN C) | +FT (BN G) | +FT (BN I) | +L2P | +FedL2P |
|---|---|---|---|---|---|---|
| **1000** | CIFAR-10-C | 59.58±0.03 | 57.03±0.08 | 55.77±0.12 | 58.80±0.13 | **59.97±0.22** |
| ($\downarrow$ heterogeneity) | Caltech-10 | 80.97±0.33 | 36.02±25.21 | 81.43±2.16 | 75.52±3.83 | **88.85±0.89** |
| | DomainNet | 52.17±1.55 | 30.55±1.07 | 50.47±0.88 | 45.04±1.56 | **54.38±0.45** |
| **1.0** | CIFAR-10-C | 67.37±0.08 | 66.45±0.03 | 63.56±0.07 | 66.63±0.09 | **68.83±0.15** |
| | DomainNet | 62.27±0.58 | 44.15±0.11 | 59.4±0.7 | 54.75±0.12 | **63.77±0.44** |
| **0.5** | CIFAR-10-C | 74.92±0.08 | 75.24±0.17 | 71.38±0.01 | 74.48±0.06 | **76.78±0.22** |
| | DomainNet | 71.39±0.97 | 49.81±1.98 | 68.94±0.71 | 66.38±0.78 | **72.64±0.3** |
| **0.1** | CIFAR-10-C | 87.25±0.06 | 88.5±0.02 | 83.93±0.04 | 87.93±0.31 | **89.23±0.15** |
| ($\uparrow$ heterogeneity) | DomainNet | 86.03±0.47 | 69.41±1.95 | 85.35±1.14 | 83.93±1.02 | **86.36±0.45** |

to the same set of clients via fine-tuning. To this end, given the CIFAR-10 dataset partitioned among a group of clients, we pretrained $\theta_g$ following best practices from [23] using FedAvg and finetune it on the same set of clients. Table 2 shows the personalized accuracy of the various fine-tuning baselines (Section 4.1.2) and FedL2P using $e = 5\&15$ local epochs on groups of clients with varying label distribution, $P_i(y)$; $\alpha = 1000$ represents the IID case and $\alpha = 1.0, 0.5, 0.1$ represents more heterogeneous case. As observed in many previous works [48, 27], increasing label heterogeneity would result in a better initial global model at a expense of personalized performance. Our method instead retains the initial global performance and focuses on improving personalized performance.

We also show that in many cases, especially for clients with limited local compute budget $e = 5$, utilizing the pretrained model's BN statistics result (**BN G**) can be more beneficial as CIFAR-10 consists of images from the same natural image domain; in contrast, previous works mainly use either the client's BN statistics (**BN C**) or the incoming feature batch statistics (**BN I**) to normalize the features. This strategy is discovered by FedL2P, as illustrated in Fig. 2a where the learned $\boldsymbol{\beta}$ is 0 for all BN layers of all clients. For the IID case in particular, FedL2P learns a sparsity[6] of $1.0$, learning rate $\eta = 0$, for all layers in all clients, forgoing fine-tuning and using the initial global model as the personalized model. For $\alpha = 1.0$ and $0.1$, FedL2P learns highly sparse models similar to recent works that proposed fine-tuning only a subset of hand-picked layers [48, 17, 11, 2, 37] to obtain performance gains. Lastly, L2P performs worse than some standard fine-tuning baselines as it meta-overfits on each client's validation set, highlighting the benefits of FL over local HPO.

**FedL2P's Complementability with previous FL works.** As our proposed FedL2P learns to improve the FT process, it is complementary, not competing, with other FL methods that learn shared model(s). Hence, besides FedAvg, we utilize FedL2P to better personalize $\theta_g$ pretrained using PerFedAvg(HF) [15] and FedBABU [48] as shown in Table. 1, where we compare FedL2P against the most commonly used FT approach, **BN C**. Our results show that applying FedL2P to all three FL methods can lead to further gains, in most cases outperforming FT in each respective method. This performance improvement can also bridge the performance gap between different methods. For instance, while FedAvg+FT has worse performance than FedBABU+FT in all cases, FedAvg+FedL2P obtained comparable or better performance than FedBABU+FT for $\alpha = 1000 \& 0.1$.

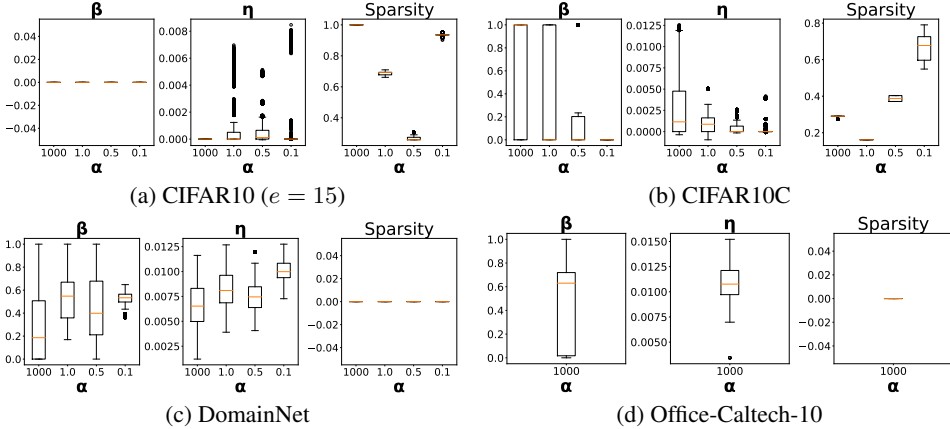

Figure 2: Locality, spread, and skewness of FedL2P's learned hyperparameters, $\boldsymbol{\beta}$ and $\boldsymbol{\eta}$, of different layers across clients and the model's sparsity of all clients for each personalization scenario.

### 4.3 Personalizing to Unseen Clients

**Unseen during Pretraining.** We evaluate the performance of FedL2P on CIFAR-10-C starting from the pretrained model trained using FedAvg on CIFAR-10 in the IID setting $\alpha = 1000$ (Section. 4.2) as

---

[6]Sparsity refers to the percent of parameters whose learned learning rate for FT is 0.

Table 4: Experiments on Speech Commands. Both the pretrained model and meta-nets are only learned on the *seen* pool of clients.

| Client Pool | Epochs (e) | Global Accuracy | +FT (BN C) | +FT (BN G) | +FT (BN I) | +L2P | +FedL2P |
|---|---|---|---|---|---|---|---|
| *Seen* | 5 | 82.11 | 70.07±0.63 | 2.98±0.00 | 65.97±0.43 | 64.79±1.84 | **87.77±0.47** |
| | 15 | 82.11 | 74.15±0.70 | 2.98±0.0 | 68.98±0.19 | 65.80±1.66 | **84.74±0.38** |
| *Unseen* | 5 | 81.62 | 62.76±1.83 | 2.85±0.00 | 64.88±0.66 | - | **87.85±0.18** |
| | 15 | 81.62 | 68.76±3.40 | 2.85±0.00 | 67.28±0.46 | - | **84.60±0.35** |

shown in Table. 3. These new clients, whose data is partitioned from CIFAR-10-C, did not participate in the pretraining of the global model and, instead, only participate in the training of the meta-nets. As added noise would impact the feature shift among clients, Fig. 2b shows FedL2P learns to use $\beta = 1$ as compared to sole use of $\beta = 0$ in the CIFAR-10 case (Fig. 2a); some clients benefit more by using its own BN statistics. Hence, the performance gap for non-IID cases between **BN C** and **BN G** is significantly smaller as compared to CIFAR-10 shown in Table. 2.

**Unseen during Learning of Meta-nets.** We evaluate the case where some clients are not included in the learning of the meta-nets through our experiments on the Speech Commands dataset. We first pretrain the model using FedAvg and then train the meta-nets using our *seen* pool of clients. The learned meta-nets are then used to fine-tune and evaluate both our *seen* and *unseen* pool of clients as shown in Table. 4. Note that L2P is not evaluated on the *unseen* pool of clients as we only evaluate the meta-nets learned on the *seen* pool of clients. We see that fine-tuning the pretrained model on each client's train set resulted in a significant drop in test set performance. Moreover, as each client represents a different speaker, fine-tuning using the pretrained model's BN statistics (**BN G**) fails entirely. FedL2P, on the other hand, led to a significant performance boost on both the *seen* and *unseen* client pools, e.g. by utilizing a mix of pretrained model's BN statistics and the client's BN statistics shown in Appendix F.

Table 5: Ablation study for FedL2P with $e = 15$.

| $\alpha$ | Dataset | +FT (BN C) | +FT (BN G) | +FedL2P (BNNet) | +FedL2P (LRNet) $\beta=1$ | +FedL2P (LRNet) $\beta=0$ | +FedL2P |
|---|---|---|---|---|---|---|---|
| **1000** | CIFAR-10 | 63.04±0.02 | 59.85±0.04 | 62.35±0.24 | 62.62±0.21 | 65.11±0.02 | **65.13±0.02** |
| (↓ heterogeneity) | CIFAR-10-C | 59.58±0.03 | 57.03±0.08 | 59.57±0.13 | **60.09±0.02** | 59.30±0.11 | 59.97±0.22 |
| | Caltech-10 | 80.97±0.33 | 36.02±25.21 | 88.12±1.18 | 85.50±5.76 | 42.59±22.87 | **88.85±0.89** |
| | DomainNet | 52.17±1.55 | 30.55±1.07 | 53.39±0.85 | **55.59±2.76** | 44.43±3.46 | 54.38±0.45 |
| **1.0** | CIFAR-10 | 61.42±0.13 | 63.23±0.15 | 63.75±0.04 | 64.67±0.06 | 64.61±0.49 | **65.76±0.31** |
| | CIFAR-10-C | 67.37±0.08 | 66.45±0.03 | 68.1±0.07 | 68.62±0.07 | 67.82±0.1 | **68.83±0.15** |
| | DomainNet | 62.27±0.58 | 44.15±0.11 | 62.73±0.51 | 63.69±0.43 | diverge | **63.77±0.44** |
| **0.5** | CIFAR-10 | 62.34±0.14 | 67.4±0.06 | 67.59±0.15 | **68.81±0.05** | 68.01±0.29 | 68.45±0.50 |
| | CIFAR-10-C | 74.92±0.08 | 75.24±0.17 | 76.36±0.08 | **76.86±0.06** | 76.11±0.07 | 76.82±0.19 |
| | DomainNet | 71.39±0.97 | 49.81±1.98 | 70.99±1.15 | **72.74±0.51** | diverge | 72.64±0.30 |
| **0.1** | CIFAR-10 | 79.15±0.07 | 78.97±0.07 | 79.47±0.2 | 80.24±0.09 | **80.39±0.15** | 80.28±0.07 |
| (↑ heterogeneity) | CIFAR-10-C | 87.25±0.06 | 88.5±0.02 | 88.6±0.1 | 89.08±0.04 | 89.14±0.13 | **89.23±0.15** |
| | DomainNet | 86.03±0.47 | 69.41±1.95 | 85.87±1.31 | 85.78±0.6 | diverge | **86.36±0.45** |

## 4.4 Personalizing to Different Domains

We evaluate our method on Office-Caltech-10 and DomainNet datasets commonly used in domain generalization/adaptation, which exhibit both marginal and conditional feature shifts. Differing from the conventional FL setup, we adopted a pretrained model trained using ImageNet [13] and attached a prediction head as our global model. Similar to CIFAR-10-C experiments, we compare our method with the base-

Table 6: ARI values between the clustering of clients by their domains and inputs/outputs of the BNNet and LRNet.

| Dataset | $\alpha$ | BNNet | | LRNet | |
|---|---|---|---|---|---|
| | | Input ($\xi$) | Output ($\beta$) | Input ($x$) | Output ($\eta$) |
| Caltech-10 | 1000 | 1.0 | 1.0 | 1.0 | 1.0 |
| DomainNet | 1000 | 0.65 | 0.64 | 0.76 | 0.77 |
| | 1.0 | 0.70 | 0.59 | 0.77 | 0.74 |
| | 0.5 | 0.52 | 0.53 | 0.72 | 0.63 |
| | 0.1 | 0.27 | 0.52 | 0.65 | 0.59 |

lines in Table. 3. Unlike CIFAR-10/10-C, Office-Caltech-10 and DomainNet consist of images from distinct domains and adapting to these domain result in a model sparsity of 0 as shown in Fig. 2c and Fig. 2d. Hence, a better personalization performance is achieved using the client's own local BN statistics (**BN C**) or the incoming batch statistics (**BN I**) than the pretrained model's (**BN G**). Lastly, each client's personalized model uses an extent of the pretrained natural image statistics as well as its own domain statistics, for both datasets.

**Further Analysis** To further investigate how well the hyperparameters returned by our FedL2P-trained meta-nets capture the domain-specific information, we analyze the clustering of local features, namely $\boldsymbol{\xi} = (\xi_1, \ldots, \xi_B)$ and $\boldsymbol{x} = (E(x_0), SD(x_0), \ldots, E(x_{M-1}), SD(x_{M-1}))$ (i.e., the inputs to the meta-nets), and the resulting respective hyperparameters, $\boldsymbol{\beta}$ and $\boldsymbol{\eta}$ (i.e., outputs of the meta-nets), among clients. Specifically, we compute the similarity matrix using the Euclidean distance between the features/hyperparameters of any two clients and perform spectral clustering[7], using the

---
[7]We use the default parameters for spectral clustering from `scikit-learn` [50].

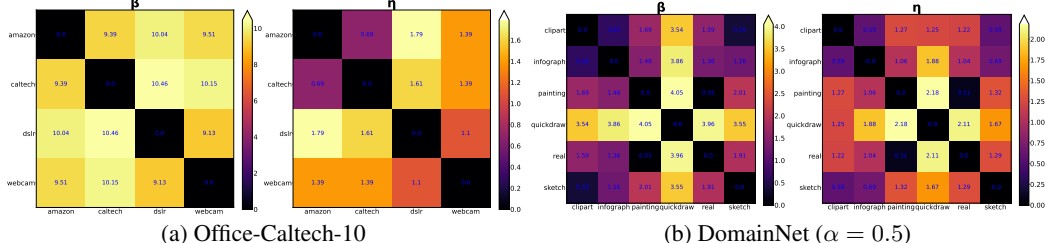

(a) Office-Caltech-10          (b) DomainNet ($\alpha = 0.5$)

Figure 3: Cluster distance maps. Each block represents *normalized* distance, where the distance of the block $(j, k)$ is measured as the average of the Euclidean distances between all pairs of clients' $\beta$ that belong to domain $j$ and domain $k$ (similarly for $\eta$'s), normalized by the within-domain distance (see text). An off-diagonal block $> 0$ indicates that the corresponding clusters are better aligned with the true domains.

discretization approach proposed in [58] to assign labels on the normalized Laplacian embedding. We then compute the *Adjusted Rand Index* (ARI) [25] between the estimated clusters and the clusters partitioned by the different data domains as shown in Table. 6. We also visualize the alignment between the estimated clusters and the true domains in Fig. 3, where each block $(j, k)$ represents the *normalized* average Euclidean distance between all pairs of clients in domain $j$ and $k$. Specifically, we divide the mean distance between domain $j$ and $k$ by the within-domain mean distances and take the log scale for better visualization: $\log(\frac{d(j,k)}{\sqrt{d(j,j)}\sqrt{d(k,k)}})$ where $d(j, k)$ is the mean Euclidean distance between $j$ and $k$. Thus, a random clustering has distance close to 0 and $> 0$ indicates better alignment between the clustering and the true domains.

As shown, for DomainNet, both BNNet and LRNet consistently preserve the cluster information found in their inputs, $\xi$ & $x$, respectively. However, perfect clustering is not achieved due to the inherent difficulty. For instance, the *real* and *painting* domains share similar features, resulting in similar hyperparameters; the cross-domain distance between *real* and *painting* is $\sim 0$ in log-scale in Fig. 3 and hence indistinguishable from their true domains. In contrast, the clients' features and resulting hyperparameters of the Office-Caltech-10 dataset are perfectly clustered (ARI=1) as visualized in Fig. 3a and Appendix. F.

### 4.5 Ablation Study

To elucidate the individual impact of BNNet & LRNet, we run an ablation study of all of the datasets used in our experiments and present the results in Table. 5, where CIFAR10 adopts the pretrained model trained using FedAvg. As BNNet learns to weight between client's BN statistics (**BN C**) and pretrained model's BN statistics (**BN G**), running FedL2P with BNNet alone leads to either better or similar performance to the better performing baseline. Running LRNet as a standalone, on the other hand, can result in further gains, surpassing the use of both BNNet and LRNet on some benchmarks. Nonetheless, it requires prior knowledge of the data feature distribution of the client in order to set a suitable $\beta$, of which $\beta = 1$ uses **BN C** and $\beta = 0$ uses **BN G**. Our approach assumes no knowledge of the client's data and learns an estimated $\beta$ per-scenario and per-client using BNNet.

## 5 Conclusion

In this paper, we propose FedL2P, a framework for federated learning of personalization strategies specific to individual FL scenarios and datasets as well as individual clients. We learned meta-nets that use clients' local data statistics relative to the pretrained model, to generate hyperparameters that explicitly target the normalization, scaling, and shifting of features as well as layer-wise parameter selection to mitigate the detrimental impacts of both marginal and conditional feature shift and marginal label shift, significantly boosting personalized performance. This framework is complementary to existing FL works that learn shared model(s) and can discover many previous hand-designed heuristics for sparse layer updates and BN parameter selection as special cases, and learns to apply them where appropriate according to the specific scenario for each specific client. As a future work, our approach can be extended to include other hyperparameters and model other forms of heterogeneity, e.g. using the number of samples as an expert input feature to a meta-net.

### Acknowledgements

This work was supported by Samsung AI and the European Research Council via the REDIAL project.

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

# Appendix

## A  Derivation of Equation (7)

From Eq (5), we know that:

$$\frac{d\mathcal{L}_{i,T}(\theta_i^*, \lambda)}{d\theta} = 0 \tag{11}$$

Based on the implicit functional theorem (IFT), we get that if we have a function $F(x,y) = c$, we can derive that $y'(x) = -F_x/F_y$. Therefore, plus the Eq (11) into the theorem, we can get:

$$\frac{d\theta^*}{d\lambda} = -\frac{\partial_\theta \left(\frac{d\mathcal{L}_{i,T}(\theta_i^*, \lambda)}{d\theta}\right)}{\partial_\lambda \frac{d\mathcal{L}_{i,T}(\theta_i^*, \lambda)}{d\theta}} = -(\partial_\theta^2 \mathcal{L}_T(\theta, \lambda))^{-1}\, \partial_{\lambda\theta} \mathcal{L}_T(\theta, \lambda) \tag{12}$$

## B  Positioning of FedL2P.

Table 7 shows the positioning of FedL2P against existing literature. Note that this list is by no means exhaustive but representative to highlight the position of our work. Most existing approaches obtained personalized models using a personalized policy and local data, often through a finetuning-based approach. This personalized policy can either 1) be fixed, *e.g.* hand-crafting hyperparameters, layers to freeze, selecting number of mixture components, number of clusters or 2) learned, *e.g.* learning a hypernetwork to generate weights or meta-nets to generate hyperparameters. These approaches are also grouped based on whether this personalized policy is dependent on the local data during inference, *e.g.* meta-nets require local client meta-data to generate hyperparameters.

In order to adapt to per-dataset per-client scenarios, many works rely on storing per-client personalized layers, which are trained only on each client's local data. Unfortunately, the memory cost of storing these models scales with the number of clients, C, restricting previous works to small scale experiments. We show that these works are impractical in our CIFAR-10 setup of 1000 clients in Appendix. C. Moreover, most existing methods rely on a fixed personalized policy, such as deriving shared global hyperparameters for all clients in FLoRA, or they do not dependent on local data, such as FedEx which randomly samples per-client hyperparameters from learned categorical distributions. Hence, these methods do not adapt as well to per-dataset per-client scenarios and are ineffective at targeting both label and feature shifts. Lastly, although pFedHN and pFedLA target both label and feature shift cases, they do not scale well to our experiments as shown in Appendix. C.

Table 7: Positioning of FedL2P with existing FL approaches. C is the total number of clients, M is the number of layers in the model, B is the number of BN layers in the model, D is the number of data domains, H is the number of hidden layers in the hypernetwork.

| FL Approach | Learns Shared Model(s) | Personalized Layers | Personalized Policy Obtained by? | Personalized Policy Data Dependent? | Targets Label Shift | Targets Feature Shift | Memory Cost | Scale to Large Networks |
|---|---|---|---|---|---|---|---|---|
| FedProx [33] PerFedAvg [15] pFedMe [59] Ditto [34] MOON [32] FedBABU [48] | Yes | No | Fixed | No | Yes | No | O(M) | ✓ |
| PerFedMask [54] | Yes | No | Fixed | No | Yes | No | O(M) | ✓ |
| FedBN [35] | Yes | Yes | Fixed | No | No | Yes | O(CB) | ✓ |
| FedPer [2] FedRep [11] APFL [14] LG-FedAvg [37] IFCA [17] | Yes | Yes | Fixed | No | Yes | No | O(CM) | ✗ |
| FedDAR [64] | Yes | Yes | Fixed | No | Yes | Yes | O(DM) | ✗ |
| FLoRA [65] | Yes | No | Fixed | No | Yes | No | O(M) | ✓ |
| FedEx [30] | Yes | Supported | Learned | No | Yes | No | O(M) | ✓ |
| FedEM [43] | Yes | No | Fixed | No | Yes | No | O(M) | ✓ |
| FedFOMO [62] FedMe [44] | No | Yes | Fixed | No | Yes | No | O(CM) | ✗ |
| pFedHN [55] pFedLA [40] | No | Yes | Learned | Yes | Yes | Yes | O(CMH) | ✗ |
| FedL2P (Ours) | No | Supported | Learned | Yes | Yes | Yes | O(M) | ✓ |

Most importantly, all existing FL approaches shown in Table 7 can use finetuning either to personalize shared global model(s) or as a complementary personalization strategy to further adapt their personalized models. Since our proposed FedL2P focus on better personalizing shared global model(s) by learning better a personalized policy which leverages the clients' local data statistics relative to the given pretrain model(s), our approach is complementary to all existing FL solutions that learn shared model(s); we showed improvements over a few of these works in Table 1.

**Use Cases of FedL2P** Mainstream FL focuses on training from scratch, but we focus on federated learning of a strategy to adapt an existing pre-trained model (whether obtained by FL or not) on an unseen group of clients with heterogeneous data. There are several scenarios where this setup and our solution would be useful:

1. Scenarios where it's expensive to train from scratch for a new group of clients, e.g. adopting FedEx [30] from scratch for a new group of clients would require thousands of rounds to retrain the model and HP while our method takes hundreds to learn two tiny meta-nets (Appendix. C).

2. Scenarios where there is a publicly available pre-trained foundation model that can be exploited. This is illustrated in Section 4.4 where we adapt a publicly available pretrained model trained using ImageNet on domain generalization datasets.

3. Scenarios where it's important to also maintain a global model with high initial accuracy - often neglected by previous personalized FL works.

Note that our approach also does not critically depend on the global model's performance. Even in the worst case where the input statistics derived from the global model are junk (e.g., they degenerate to a constant vector, or are simply a random noise vector), then it just means the hyperparameters learned are no longer input-dependent. In this case, FedL2P would effectively learn a constant vector of layer-wise learning rates + BN mixing ratio, as opposed to a function that predicts them. Thus, in this worst case we would lose the ability to customize the hyperparameters differently to different heterogeneous clients, but we would still be better off than the standard approach where these optimization hyperparameters are not learned. In the case where our global-model derived input features are better than this degenerate worst case, FedL2P's meta-nets will improve on this already strong starting point.

# C   Cost of FedL2P

**Computational Cost of Hessian Approximation.** We compare with hessian-free approaches, namely first-order (FO) MAML and hessian-free (HF) MAML, both of which are used by PerFedAvg, and measure the time it took to compute the meta-gradient after fine-tuning. Specifically, we run 100 iterations of each algorithm and report the mean of the walltime. Our proposed method takes 0.24 seconds to compute the hypergradient, 0.12 seconds of which is used to approximate the Hessian. In comparison, FO-MAML took 0.08 seconds and HF-MAML took 0.16 seconds to compute the meta-gradient. Hence, our proposed method is not a significant overhead relative to simpler non-Hessian methods. It is also worth noting that the number of fine-tuning epochs would not impact the cost of computing the hypergradient.

**Memory Cost.** In our CIFAR10 experiments, the meta-update of FedL2P has a peak memory usage of 1.3GB. In contrast, existing FL methods that generate personalized policies require an order(s) of magnitude more memory and hence only evaluated in relatively small setups with smaller networks. For instance, pFedHN [55] requires in a peak memory usage of 17.93GB in our CIFAR10 setup as its user embeddings and hypernetwork scale up with the number of clients and model size. Moreover, they fail to generate reasonable client weights as these techniques do not scale to larger ResNets used in our experiments. APFL [14], on the other hand, requires each client to maintain three models: local, global, and mixed personalized. Adopting APFL in our CIFAR10 setup of 1000 clients requires over 134GB of memory just to store the models per experiment, which is infeasible.

**Communication Cost.** For each FL round, we transmit the parameters of the meta-nets, which are lightweight MLP networks to from server to client and vice versa. Note that transmitting the global pretrained model to each new client is a one-time cost. Office-Caltech-10, DomainNet, and Speech Commands setups take a maximum of 100 communication rounds, 0.24% of the pretraining cost, to learn the meta-nets. CIFAR-10 and CIFAR-10-C, on the other hand, can take hundreds of

rounds up to a maximum of 500 rounds, 0.38% of the pretraining cost. In contrast, joint model and hyperparameter optimization typically takes thousands of rounds [30], having to transmit both the model and the hyperparameter distribution across the network. In summary, FedL2P incurs <1% additional costs on top of pretraining and forgoes the cost of federatedly learning a model from scratch, which can be advantageous in certain scenarios as listed in Section. B.

**Inference Cost.** During the fine-tuning stage, given the learned meta-nets, FedL2P requires 2 forward pass of the model per image and one forward pass of each meta-net to compute the personalized hyperparameters. This equates to 0.55GFLOPs per image and would incur a minor additional cost of 4.4% more than the regular finetuning process of 15 finetune epochs.

## D  Pretrained Model and Setup Details

We use the Flower federated learning framework [6] and 8 NVIDIA GeForce RTX 2080 Ti GPUs for all experiments. ResNet-18 [20] is adopted with minor differences in the various setups:

**CIFAR-10.** We replaced the first convolution with a smaller convolution $3 \times 3$ kernel with stride$= 1$ and padding$= 1$ instead of the regular $7 \times 7$ kernel. We also replaced the max pooling operation with the identity operation and set the number of output features of the last fully connected layer to 10. The model is pretrained in a federated manner using FedAvg [46] or FedBABU [48] or PerFedAvg(HF) [15] with a starting learning rate of 0.1 for 500 communication rounds. For PerFedAvg, we adopted the recommended hyperparameters used by the authors to meta-train the model. The fraction ratio is set to $r = 0.1$; 100 clients, each of who perform a single epoch update on its own local dataset before sending the updated model back to the server, participate per round. We dropped the learning rate by a factor of 0.1 at round 250 and 375. This process is repeated for each $\alpha = 1000, 1.0, 0.5, 0.1$, resulting in a pretrained model for each group of clients. We experiment with various fine-tuning learning rates $\{1.0, 0.1, 0.01, 1e-3, 1e-4, 1e-5\}$ and pick the best-performing one, $1e-3$ for all experiments; the initial value of $\tilde{\eta}$ in FedL2P is also set at $1e-3$.

**CIFAR-10-C.** We adopted the pretrained model trained in CIFAR-10 for $\alpha = 1000$ and used the same fine-tuning learning rate for all experiments.

**Office-Caltech-10 & DomainNet.** We adopted a Resnet-18 model that was pretrained on ImageNet [13] and provided by torchvision [41]. We replaced the number of output features of the last fully connected layer to 10. Similar to CIFAR-10 setup, we experiment with the same set of learning rates and pick the best-performing one, $1e-2$ for our experiments.

**Speech Commands.** We adopted the setup and hyperparameters from ZeroFL [52]; a Resnet-18 model is trained using FedAvg for 500 rounds using a starting learning rate of 0.1 and an exponential learning rate decay schedule with a base learning rate of 0.01. We use the base learning rate for fine-tuning for all experiments.

## E  Architecture & Initialization Details

We present the architecture of our proposed meta-nets, BNNet and LRNet. Both networks are 3-layer MLP models with 100 hidden layer neurons and ReLU [1] activations in-between layers. BNNet and LRNet clamp the output to a value of $[0, 1]$ and $[0, 1000]$ respectively and use a straight-through estimator [4] (STE) to propagate gradients. We also tried using a sigmoid function for BNNet which converges to the same solution but at a much slower pace. We initialize the weights of BNNet and LRNet with Xavier initialization [18] using the normal distribution with a gain value of 0.1. To control the starting initial value of BNNet and LRNet, we initialize the biases of BNNet and LRNet with constants 0.5 and 1.0, resulting in initial values of $\sim 0.5$ and $\sim 1.0$ respectively. We also tried experimenting BNNet with different initializations by setting the biases to $[0.2, 0.5, 0.8]$ and got similar results.

## F  Additional Results

**Relative Clustered Distance Maps**. We present an extension of Fig. 3 for both the inputs, $\xi$ & $x$, and outputs, $\beta$ & $\eta$, of the meta-nets in Fig. 5.

Table 8: Comparison between using FOMAML+ and IFT in FedL2P for Office-Caltech-10 and Domainnet ($e = 15$).

| $\alpha$ | Dataset | +FT (BN C) | +FedL2P (FOMAML+) | +FedL2P (IFT) |
|---|---|---|---|---|
| **1000** | Caltech-10 | 80.97±0.33 | 83.20±1.92 | **88.85±0.89** |
| | DomainNet | 52.17±1.55 | 52.70±0.17 | **54.38±0.45** |
| **1.0** | DomainNet | 62.27±0.58 | 63.14±0.38 | **63.77±0.44** |
| **0.5** | DomainNet | 71.39±0.97 | 71.37±0.79 | **72.64±0.3** |
| **0.1** | DomainNet | 86.03±0.47 | 86.22±0.16 | **86.36±0.45** |

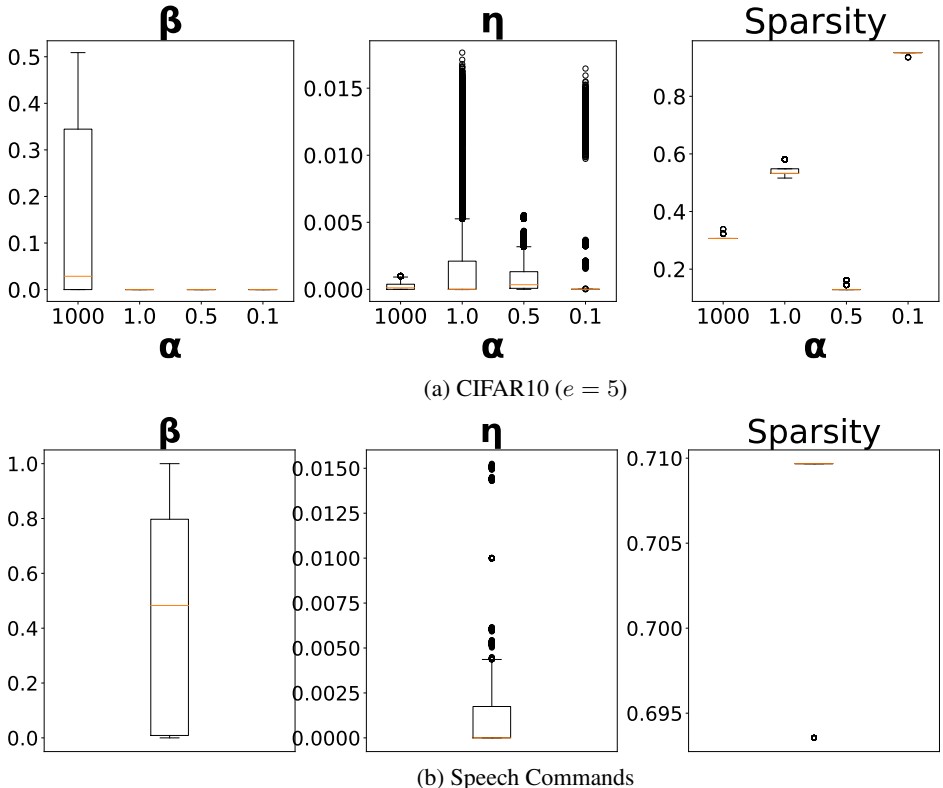

Figure 4: Locality, spread, and skewness of FedL2P's learned hyperparameters, $\beta$ and $\eta$, of different layers across clients and the model's sparsity of all clients for each personalization scenario.

**Learned Personalized Hyperparameters**. We present the learned hyperparameters for the other client groups not shown in Fig. 2 in Fig. 4.

**Comparison with FOMAML**. We remark that off-the-shelf FOMAML [16] (as an initial condition learner) is not a meaningful point of comparison because our problem is to fine-tune/personalize pre-trained models. Therefore, to compare with FOMAML, we focus on learning our FedL2P meta-nets with FOMAML style meta-gradients instead of IFT meta-gradients. In order to apply FOMAML to learning rate optimization, this also required extending FOMAML with the same trick as we did for our IFT approach as shown in Eq. 9 & 10. However, FOMAML ignores the learning trajectory except the last step, which may result in performance degradation over longer horizons. We term this baseline FOMAML+. Table. 8 reports results on the multi-domain datasets as described in Section. 4.4, keeping the same initial condition and meta representation (LRNet and BNNet meta-nets), and varying only the optimization algorithm. Our approach of using IFT to compute the best response Jacobian, outperforms FOMAML+ for $\alpha = 1000, 1.0, 0.5$ and has comparable performance for $\alpha = 0.1$.

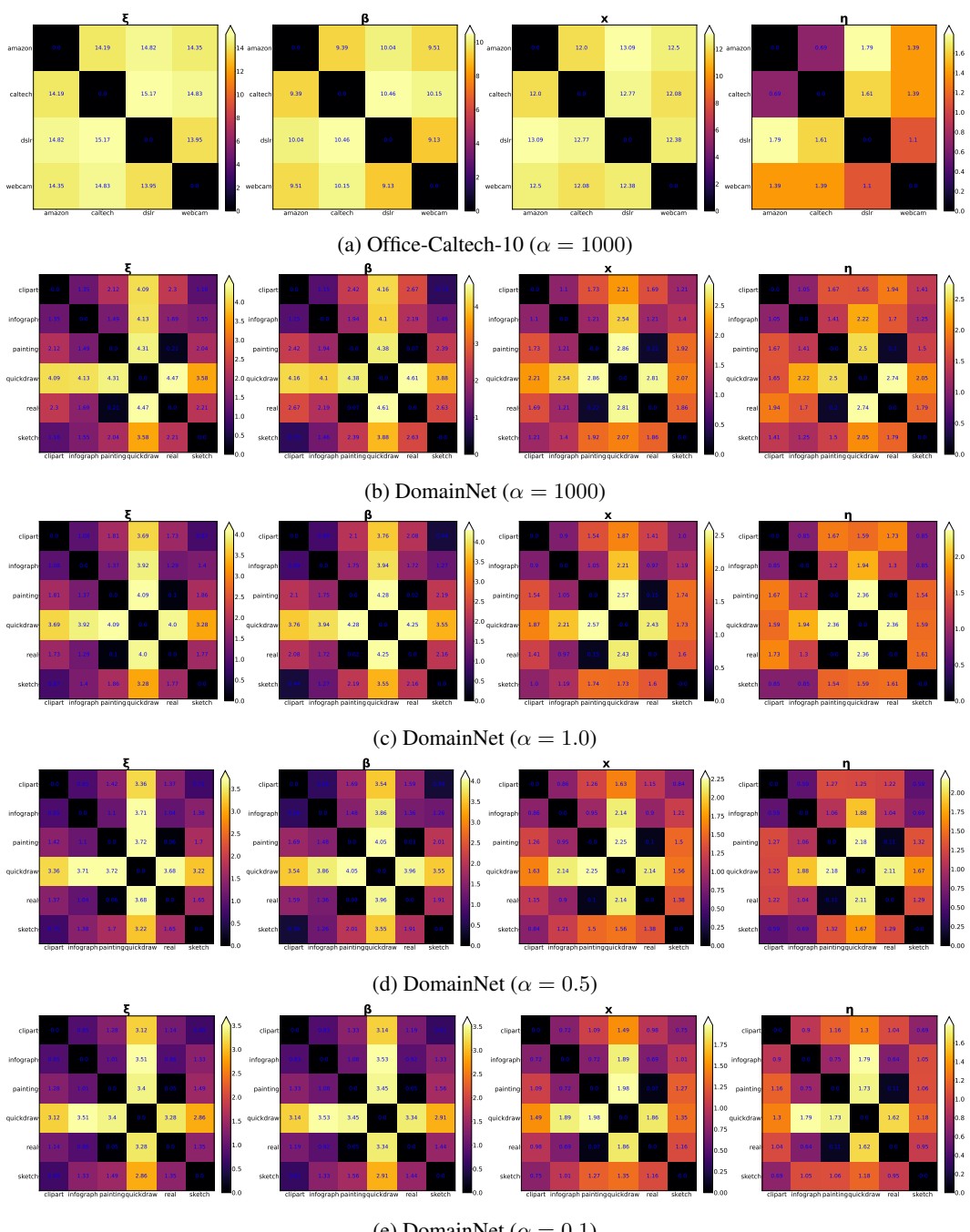

(a) Office-Caltech-10 ($\alpha = 1000$)

(b) DomainNet ($\alpha = 1000$)

(c) DomainNet ($\alpha = 1.0$)

(d) DomainNet ($\alpha = 0.5$)

(e) DomainNet ($\alpha = 0.1$)

Figure 5: Cluster distance maps. Each block represents *normalized* distance between two domains (e.g., Caltech vs. DSLR), where the distance of the block $(j, k)$ is measured as the average of the Euclidean distances between all pairs of clients' $\boldsymbol{\beta}$ that belong to domain $j$ and domain $k$ (similarly for $\boldsymbol{\eta}$'s). We normalize distance by the within-domain distances (see text), and take $\log$ for better visualization. Hence, an off-diagonal block greater than $0$ indicates that the corresponding clusters are better aligned with the true domains.

