# OpenReview forum: "FedL2P: Federated Learning to Personalize"
_NeurIPS.cc/2023/Conference — NeurIPS 2023 poster_

### Official Review · Reviewer_Jjav · 2023-06-18

**Soundness:** 3 good
**Presentation:** 3 good
**Contribution:** 3 good
**Rating:** 5
**Confidence:** 3

**Summary:**


This study primarily concentrates on refining hyperparameters in federated learning, especially concerning non-IID data. The researchers highlight two key hyperparameter groups—Batch Normalization (BN) and selective update hyperparameters—that yield impressive outcomes.

BN hyperparameters help mitigate feature shifts, a prevalent problem in federated learning. FedBN, a cited method, aids in managing feature shift more effectively by localizing the BN layers while keeping other layers global. Details of the BN layer formation and its constituents are also given.

Moreover, the study explores a global model's application, initially trained through existing federated algorithms, which is then fine-tuned to the client set. The CIFAR-10 dataset split among various clients is utilized for this process, demonstrating that their approach maintains global performance while enhancing personalized performance.

The researchers also incorporate the Jacobian product to approximate the Hessian inverse and calculate the hyper gradient, forming a part of their suggested algorithm.

For experimental purposes, ResNet-18 and SGD are consistently employed for all tests. Batch size, local epoch number, learning rate, hyper gradient clipping, maximum communication rounds, and other parameters are also predefined.

This paper contributes to federated learning by introducing a method to optimize hyperparameters, especially in non-IID data scenarios, and backs its effectiveness through empirical results.

**Strengths:**

**Originality:** The paper stands out due to its unique approach to handling non-IID data within federated learning. Implementing Batch Normalization (BN) hyperparameters to tackle feature shift is an inventive fusion of existing deep learning knowledge and federated learning's distinct hurdles. Moreover, the employment of the Jacobian product to estimate the Hessian inverse and compute the hyper gradient is a distinct application within the federated learning sphere, marking an inventive merge of established concepts to address a recurring federated learning issue.

**Quality:** The paper's quality appears robust. It reflects the authors' thoughtful research methodology and the theoretical integrity of their approach to hyperparameter optimization. Leveraging the CIFAR-10 dataset for validation is a norm in this field, and the authors thoroughly outline their experimental setup. Nevertheless, a broader evaluation across diverse datasets and juxtaposition with other methodologies could further enrich the paper's quality.

**Clarity:** The paper is well-crafted and clearly depicts the research. The authors present exhaustive descriptions of their methodology and the theoretical underpinnings of their strategy. They also lucidly detail their experimental setup and findings. Such explicitness in communication renders the paper comprehensible to a broad audience, from novices in federated learning to seasoned researchers.

**Significance:** The paper holds substantial significance. It addresses pivotal challenges, such as dealing with non-IID data and optimizing hyperparameters, vital in federated learning. The authors' solutions to these challenges could potentially lead to the more efficient and practical training of federated learning models, making a notable contribution to the field. Additionally, this paper's insights could stimulate new research pathways and methodologies within the broader realm of machine learning.

**Weaknesses:**

In summary, while the paper is fundamentally and conceptually solid, there are a few weaknesses that weaken its contribution and soundness.

1. **Dataset and Model Constraints:** The study principally uses the CIFAR-10, Office-Caltech-10, and DomainNet datasets in its experiments. Although these are recognized standards, the findings may not extend to different datasets and models. Testing the proposed method across diverse datasets (such as images, text, and tabular data) and various model architectures could better showcase the method's adaptability and resilience.
2. **Comparison to Other Techniques:** A broader comparison with cutting-edge methods could enhance the paper. While some algorithms are referenced, the relative performance of the authors' method remains unclear. A side-by-side comparison could clarify the comparative advantages and disadvantages of their approach.
3. **Computational Overhead:** The authors suggest the Jacobian product for approximating the Hessian inverse and computing the hypergradient, a fresh perspective. Appendix C mentions that the computational cost compared to FO / HF-MAML is still relatively large.
4. **Applicability:** The paper's significant contribution to federated learning cannot be overlooked, yet the universal applicability of their method warrants examination. Federated learning is applied across various sectors, each with unique data distributions, privacy stipulations, and computational resources. Discussing the limitations and potential solutions of their method would fortify the paper.
5. **Reproduction**: The codes are not available yet.

**Questions:**

1. Could you elaborate on how generalizable your method is to different federated learning scenarios? For instance, how would your method perform with different data distributions, privacy requirements, and computational resources?
2. Could you provide a more comprehensive comparison with other state-of-the-art methods? It would be helpful to understand how your method performs relative to these other approaches in terms of accuracy and computational efficiency. I am sill unclear about whether your baselines cover enough range of comparable works.
3. What particular `pre-trained` models did you use in your experiments?
4. Additional Datasets and Models: Have you considered testing your method on various datasets with different characteristics (e.g., image, text, tabular data) and different model architectures?
5. Could you discuss the limitations of your approach and potential ways to overcome these limitations? This would provide a more balanced view of your work and could help guide future research in this area.

**Limitations:**

In terms of broader societal impacts, it's not clear from the provided excerpts whether the authors have considered this aspect.

---

> ### Author Rebuttal · Authors · 2023-08-07
>
> We thank the reviewer for the detailed constructive feedback. We further evaluate our approach on a speech dataset, clarify our setups, discuss the position, costs and limitations of our approach.
>
> > Q4.1) Generalizability of our method to different FL scenarios
>
> Our method can handle both label p(y) and feature distribution p(x|y) shifts as defined in L100-105 (eg: distribution shift between amazon product & home webcam images in the Caltech-10 benchmark). Through our experiments, we show that our approach adapts well to various degrees of data heterogeneity by learning which layers to freeze or personalize and determine the extent in which to utilize the client’s or pretrained model’s data statistics, both of which were hand-designed in previous works. Hence, our approach enables per-scenario & per-client personalization learning. Please refer to Appendix B for a summary of our work in comparison with existing FL works. Regarding privacy, our method fulfills the same privacy requirements as FedAvg as we used FedAvg to train our meta-nets. Lastly, we evaluate FedL2P using both 5 & 15 finetune epochs, reflecting different computational budgets, as shown in Table 2 and Table R1 (Rebuttal PDF).
>
> > Q4.2) Could you provide a more comprehensive comparison with other state-of-the-art methods?
>
> In terms of accuracy, recent comprehensive personalized FL benchmark papers [9,43] have shown that no existing method consistently outperforms other methods in all datasets & metrics. Notably, FedAvg+FT is a high-performing baseline, even achieving state-of-the-art performance in some scenarios. Moreover, these benchmark papers also show that local fine-tuning further improves personalized accuracy for most cases, when applied across a wide range of personalized FL works. As our work aims to improve the fine-tuning process, it is directly applicable to most existing works that aim to learn shared model(s) and we showed a few examples in Table 1.
>
> In terms of efficiency, in Appendix B, we highlight that our approach only scales linearly with the no. of layers of the pretrained model. Previous works that require the storing of local layers [2,11,14] for each client do not scale as well, e.g. limiting the number of clients in their CIFAR10 experiments to 100 while we evaluate our approach on 1000 clients. In Appendix C, we quantify the memory and compute cost by comparing with a few of these existing works. Notably, we mentioned that the compute cost to compute our hypergradient is **not** a significant overhead compared to FOMAML and HF-MAML. Additionally, the cost of our approach during fine-tuning is only 4.4% more than the standard fine-tuning. Lastly, the communication cost needed to train the meta-nets is 0.38% and 0.24% of the cost needed to pretrain the model in our CIFAR10 and Speech Commands setup respectively.
>
> > Q4.3) Pretrained models used in experiments
>
> Section 4.2 (CIFAR10) follows a conventional FL setup where the model is pretrained using an existing FL approach - we showed three approaches: FedAvg, PerFedAvg, and FedBABU.
>
> Section 4.3 (CIFAR10-C) uses the model pretrained using FedAvg on $\alpha=1000$ CIFAR10 clients.
>
> Section 4.4 (Caltech-10 and DomainNet) uses a publicly available model pretrained on ImageNet, illustrating the scenario where FedL2P can also be utilized to federatedly learn how to fine-tune a publicly available model.
>
> > Q4.4) Additional Datasets and Models
>
> We have added a new evaluation on the non-IID Speech Commands V2 dataset[a]. We use the 12-class version that is naturally partitioned by speaker, with one client per speaker. It is naturally imbalanced in skew and frequency, with 2112, 256, 250 clients/speakers for train, val, and test.
>
> We randomly sampled 250 of 2112 training clients for our seen pool of clients and sampled 50 out of 256+250 validation & test clients for our unseen client pool. Each client’s data is then split 80%/20% for train & test sets, with a further 20% of the resulting train set held out to form a validation set. The pool of 250 seen clients then follows the conventional FL setup described in Sec 4.2: they both pretrain the FL model and subsequently learn the meta-nets. For pretraining, we adopt the setup & hyperparameters from ZeroFL[b], which is trained using FedAvg for 500 rounds. We then use the seen pool of clients to learn the meta-nets for an additional 100 rounds. The learned meta-nets are then used to finetune and evaluate both our seen and unseen pool of clients.
>
> In Table R1 (rebuttal PDF), we compare with standard finetuning baselines and our method without FL (L2P) for 5 or 15 epochs as described in Sec 4.1.2. Note that L2P is not evaluated on the unseen pool of clients as we only evaluate the meta-nets learned on the seen pool of clients. We see that finetuning the pretrained model on each client’s train set resulted in a significant drop in test set performance. Moreover, as each client represents a different speaker, fine-tuning using the pretrained model’s BN statistics (FT (BN G)) fails entirely. Our FedL2P, on the other hand, led to a significant performance boost on both the seen and unseen client pools, e.g. by utilizing a mix of pretrained model’s & the client’s BN statistics as seen in Fig. R1.
>
> [a] Speech Commands: A Dataset for Limited-Vocabulary Speech Recognition, arXiv 2018
>
> [b] ZeroFL: Efficient On-Device Training for Federated Learning With Local Sparsity, ICLR 2022
>
>
> > Q4.5) Limitations and potential solutions
>
> We select hyperparameters that are previously shown to be critical to performance - learning rate and batch norm statistics. As mentioned by the reviewer, discrete hyperparameters, such as batch size & local epoch are manually defined. Optimizing discrete hyperparameters are outside of our scope. A possible extension of our work could include these hyperparameters by defining them as categorical distributions. Another limitation is our approach requires additional rounds to learn the meta-nets.

---

> > ### Comment · Reviewer_Jjav · 2023-08-11
> >
> > Thanks for your detailed rebuttal.

---

> > > ### Author Response · Authors · 2023-08-14
> > >
> > > Thanks for the feedback. If you are satisfied by our responses, please consider increasing your score.
> > > Otherwise, please let us know any remaining concerns you have so we can try to answer them.

---

### Official Review · Reviewer_wFN9 · 2023-07-06

**Soundness:** 2 fair
**Presentation:** 2 fair
**Contribution:** 3 good
**Rating:** 6
**Confidence:** 3

**Summary:**

This paper proposes a new personalization strategy for the Federated Learning setting. Their key idea is to train a meta-network that takes as input the client's data and outputs optimal hyperparameters for that data. These optimal hyperparameters can then be used to personalize the pretrained model. The hyperparameters they choose to output control the batch-norm mean, std-dev and per layer learning rate. They evaluate their technique on 4 datasets, and show that their technique outperforms other personalization strategies on these datasets.

**Strengths:**

- The core idea of using a meta-net proposed by the authors is quite novel and I believe it is a very promising direction.
- The hypergradient technique they propose to train this metanet is also quite novel and interesting.
- For the most part their algorithm and technique is clearly presented.

**Weaknesses:**

- The experiment setup is not clearly defined and confusing. Examples
   - In their CIFAR10 experiments, they say 20% of the data is used for test, it's not clear if that means 20% of the clients are not used in the FL step or 20% of the data in each client is not used in the FL step.
  - It is not clear what fraction of clients are used for training the FL model in each expt
  - Figures 2 and 3 are poorly labeled and not clearly explained. Figure 3 is completely unreadable.

- Experimental results are not very strong
  - FedL2P seems very computationally expensive (total compute used is not compared anywhere) and the results are only marginally better than other techniques.
  - No evaluation on actual non-IID datasets. There are several non-IID datasets available publicly for FL - EMNIST, Shakespeare, StackOverflow etc. Evaluating their method on these datasets would make the results much more compelling.
 - No evaluation on non-image datasets.

- Hyperparameters limited to continuous values - many hyperparameters like # epochs, choice of optimizer etc. are not continuous.

**Questions:**

N/A

---

> ### Author Rebuttal · Authors · 2023-08-08
>
> We thank the reviewer for their valuable feedback. We provide clarifications and evaluate on a speech dataset, including the case where some clients are not used in learning the meta-nets.
>
> > Q3.1) Clarify client/data split.
>
> In our CIFAR10 experiments, 20% of the train data is used to construct the val set. The resulting train set is used to pretrain the model in a federated setting and both the train and val sets are used to train the meta-nets given the pretrained model.
>
> > Q3.2) Fraction of clients used for training the FL model in each exp?
>
> CIFAR10 (Sec 4.2): all clients are used to pretrain the FL model and the same set of clients is subsequently re-used to learn the meta-nets - this is based on the conventional FL setup of learning global model(s) and then personalizing to each client via finetuning.
>
> CIFAR10-C (Sec 4.3): we used the pretrained model trained using CIFAR10 clients in Sec 4.2 and learned the meta-nets using all CIFAR10-C clients. Hence, the meta-nets are learnt on data which is not seen during the pretraining stage.
>
> OfficeCaltech/DomainNet (Sec 4.4): we used a publicly available ImageNet pretrained model and learned the meta-nets using all clients, showcasing FedL2P’s ability to adapt to different image domains using existing models which may not be pretrained in a FL setting.
>
> Upon reading the reviewer’s comments, we are inspired and included the case where some clients are not included in the learning of the meta-nets in Q3.4.
>
> > Q3.3) Fig. 2 & 3 are unclear.
>
> Sorry for the lack of clarity. The figures should be interpreted with the text in L317-353. We will revise the figures to make them clearer and self-contained.
>
> To summarize the findings:
>
> **Fig. 2** shows the outputs of the BNNet ($\beta$) LRNet ($\eta$), and the sparsity (% of params with learning rate 0), across layers and clients for $\alpha=1000,1.0,0.5,0.1$.
>
> - In CIFAR10 (Fig 2a), BNNet learns to use the pretrained model’s batch norm (BN) statistics ($\beta=0$) as images come from the same natural image domain as the images used to pretrain the model. When we adapt a natural image pretrained model to other domains, BNNet learns to utilize a mix of pretrained model’s BN statistics and the client’s BN statistics (Fig 2b,c,d). In some cases, it uses solely the client’s BN statistics ($\beta=1$). Conclusion: BNNet explicitly learns to handle the feature shift in both marginal and conditional distribution skews.
>
> - LRNet learns the layer-wise learning rate, freezing a proportion of the model indicated by the sparsity value. Notably, as standard fine-tuning led to performance degradation in the IID case for CIFAR10 (Fig 2a, $\alpha=1000$), LRNet learned to freeze the entire model ($\eta=0$) and hence skip fine-tuning entirely. In contrast, LRNet correctly finetunes the entire model (sparsity=0) when the image domain largely differs from the one used to train the pretrained model (Fig 2c,d).
>
> **Fig. 3** visualizes how close the estimated clusters, clustered by the generated $\beta$ and $\eta$, are to the true domain clusters. Each block $(j,k)$ shows the mean Euclidean distance of $\beta$ and $\eta$ between all pairs of clients in domain $j$ and $k$, normalized by the within-domain Euclidean distance, hence 0 in log-scale for diagonal blocks. If $\beta$ and $\eta$ are similar in two different domains (off-diagonal blocks), then the distance would be close to 0 and the domains are indistinguishable. Likewise, the bigger the distance, the more distinguishable the domains are to their true domains.
>
> > Q3.4) No evaluation on actual non-IID and non-image datasets
>
> We follow standard evaluation practice [23,58,50] of non-IID splitting existing datasets with controlled class proportion skew p(y), to analyze how performance varies with class heterogeneity (Dirichlet $\alpha=1000,1.0,0.5,0.1$; Tab 1-4). We already go beyond most existing studies and study two other types of heterogeneity: marginal feature p(x) and concept p(x|y) shift – as described in L102-105 using Office-Caltech and DomainNet. These challenging benchmarks are very much “real” non-IID – see their original papers for details of the substantial distribution shifts involved. Nevertheless, given the reviewer 's suggestion on non-image data, we have added a new speech dataset in our evaluation. Please kindly refer to Q4.4 under our response to Reviewer Jjav.
>
> > Q3.5) FedL2P computationally expensive + marginal performance improvements
>
> In CIFAR10/SpeechCommands, we trained our meta-nets for 500/100 rounds respectively (vs 500 rounds for pre-training in both cases). Nonetheless, the comms cost required to learn the meta-nets is only 0.38% and 0.24% of the pre-training cost respectively. Thus, there is a modest, but not prohibitive, additional compute cost required to learn the tiny meta-nets; and the comms cost, which is usually the main concern in FL, is marginal at < 1%.
>
> Furthermore, we emphasize that once the meta-nets are trained, the cost is **amortized**; we can use them for as many new unseen clients – as we demonstrated in Q3.4. In the limit of a large no. of new clients the per-client cost of meta-net training tends toward 0.
> Overall FedL2P is also memory-efficient and scalable as compared to existing works (Appendix B & C), has a non-significant overhead compared to hessian-free approaches, and has a marginal additional cost of 4.4% during fine-tuning compared to standard fine-tuning (Appendix C).
>
> We are not sure why the reviewer thinks the improvement is marginal. For instance, we extend Table 1 by listing the improvement in accuracy over the baselines (Rebuttal PDF Table R2), where the only marginal improvement is the case for FedBABU $\alpha=0.1$
>
> > Q3.6) Hyperparameters limited to continuous values
>
> We chose batch norm and layer-wise selective update hyperparameters as they have been shown [2,34,36,46] to be effective at dealing with feature and label shift respectively. Optimizing discrete hyperparameters is left for future work.

---

> > ### Comment · Reviewer_wFN9 · 2023-08-16
> > **Thanks for rebuttal**
> >
> > Thanks for the detailed rebuttal. I have updated my score, please do ensure that the final version has the changes.

---

### Official Review · Reviewer_aauE · 2023-07-10

**Soundness:** 3 good
**Presentation:** 2 fair
**Contribution:** 3 good
**Rating:** 5
**Confidence:** 3

**Summary:**

The paper introduces a novel approach that utilizes meta-learning to learn hypernets that aim to determine the optimal fine-tuning hyperparameters for individual clients. The proposed idea is intriguing and holds promise. The results presented in the paper are compelling. However, the paper would benefit from improvements in terms of writing and clarity. Overall, it is a borderline paper.

--- post rebuttal --

In the rebuttal, the author address my questions about clarity and positioning their work. So I increase my score from 4 to 5.

**Strengths:**

- The paper proposes a novel idea of using meta-learning to learn hypernet that can estimate the best fine-tuning hyperparameters for personalizing local models.
- The results show the proposed FedL2P can effectively improve performance on multiple standard FL methods and remain robust when there is a data distribution shift across clients.

**Weaknesses:**

- The **clarity** needs to be improved. Especially, some details of the proposed method are missing. Please see the question section below.
- The proposed method is **limited** to one certain type of personalized federated learning, i.e., fine-tuning local models. There are other strategies of personalization that are not applicable to this work. For example, one strategy is decoupled model (global shared encoder + local predictive head) [1,2]. Another strategy could be using Hypernet to directly optimize different local models [3].

[1] Zhong, A., He, H., Ren, Z., Li, N. and Li, Q., 2022, September. FedDAR: Federated Domain-Aware Representation Learning. In The Eleventh International Conference on Learning Representations.

[2] Collins, L., Hassani, H., Mokhtari, A. and Shakkottai, S., 2021, July. Exploiting shared representations for personalized federated learning. In International conference on machine learning (pp. 2089-2099). PMLR.

[3] Shamsian, A., Navon, A., Fetaya, E. and Chechik, G., 2021, July. Personalized federated learning using hypernetworks. In International Conference on Machine Learning (pp. 9489-9502). PMLR.

**Questions:**

- (line 184) Could you explain what is "the degree of data heterogeneity of the client’s local data with respect to the pretrained model"? The data heterogeneity comes from local data and the pretrained model?
- (line 205) Should $BNNet(w_{bn}; ξ_1, ξ_2, . . . , ξ_{B−1}, ξ_B)$ be $BNNet(ξ_1, ξ_2, . . . , ξ_{B−1}, ξ_B;w_{bn})$? It seems that  $ξ_1, ξ_2, . . . , ξ_{B−1}, ξ_B$ are BNNet's input.
- (line 206) Could you explain why there is a Hadamard product for the LRNet's output?
- title of section 3.3 is a bit weird. why you are Define Training/Validation Losses for IFT (implicit function theorem). Isn't it that training/validation losses should defined by your tasks?

**Limitations:**

Not discussed.

---

> ### Author Rebuttal · Authors · 2023-08-07
>
> Thank you for the constructive feedback and valuable time spent on our paper. We will revise the paper accordingly to address the points you have made to improve clarity.
>
> > Q2.1) The proposed method is limited to one certain type of personalized federated learning, i.e., fine-tuning local models. There are other strategies of personalization that are not applicable to this work. For example, one strategy is decoupled model (global shared encoder + local predictive head) [1,2]. Another strategy could be using Hypernet to directly optimize different local models [3].
>
> As summarized in recent FL benchmark studies ([9,43] in the paper), the majority of FL works can benefit from local fine-tuning, often leading to state-of-the-art personalized FL results in different datasets and metrics. As our approach aims to improve the fine-tune process, it is widely applicable and complementary to most non-fine-tuning approaches to personalization. This includes the works you mentioned [1,2] as demonstrated by further gains on FedBABU, which likewise learns a global shared encoder + local predictive head, as shown in Table 1. For a more comprehensive positioning of our work as compared to existing personalized FL literature, please refer to Appendix B.
>
> Unfortunately, pFedHN [3], which you mentioned, does not work in practice for larger setups. Note that the authors in [3] used a small model of 5 layers with a maximum of 100 clients as opposed to e.g. our CIFAR10 experiments, full-blown ResNet with 1000 clients. We tried comparing with pFedHN using their official codebase in our setup but the hypernet failed to scale to larger ResNets. This, along with its memory cost, is detailed in Appendix C.
>
> > Q2.2) (line 184) Could you explain what is "the degree of data heterogeneity of the client’s local data with respect to the pretrained model"? The data heterogeneity comes from local data and the pretrained model?
>
> Sorry for the confusing wording. We learn functions (meta-nets) to generate hyperparameters according to how different the client’s local data to the data that the model is pretrained on. As we do not store the pretraining data, we summarize it in terms of the mean and variance stored in the batch normalization layers of the pretrained model. Hence, there is a notion of data heterogeneity between the client’s data and the BN statistics stored in the pretrained model  (which summarize the pre-trained data).
>
> > Q2.3) (line 205) Should $\text{BNNet}(\boldsymbol{w}\_{bn};\xi\_1,\xi\_2,...,\xi\_{B-1},\xi\_{B})$ be $\text{BNNet}(\xi\_1,\xi\_2,...,\xi\_{B-1},\xi\_{B};\boldsymbol{w}\_{bn})$? It seems that $\xi\_1,\xi\_2,...,\xi\_{B-1},\xi\_{B}$ are BNNet's input.
>
> Yes. We adopted the notation of putting the parameters before the input consistently throughout the paper. If you recommend putting the input first, we are happy to change the notation in the revised paper.
>
> > Q2.4) (line 206) Could you explain why there is a Hadamard product for the LRNet's output?
>
> The learning rate used for fine-tuning is a product of the output of the LRNet and the learnable post multiplier parameter (Eq. 4). This parameter is the starting learning rate, initialized with the best learning-rate from our grid search prior to meta-training (Appendix E). Having this scaling parameter fixed would limit the range of the resulting learning rate as the LRNet has to learn the optimal learning-rate given a fixed number of optimization steps. Making this parameter learnable would better control the potential effect this parameter could have on the learning rate.
>
> > Q2.5) title of section 3.3 is a bit weird. why you are Define Training/Validation Losses for IFT (implicit function theorem). Isn't it that training/validation losses should defined by your tasks?
>
> Yes, we agree that the training and validation losses are defined by the task (e.g. cross entropy loss on a classification dataset). However, we had to explicitly modify this loss to make the loss dependent on the learning rate as shown in Eq. 9 & 10, in order to be able to optimize it with IFT. We will revise the title of Section 3.3 to “Adapting the Losses for IFT”.

---

> > ### Comment · Reviewer_aauE · 2023-08-14
> > **Thanks for your reply**
> >
> > The author did a greate job in the rebuttal. Most of my questions are answered. I will increase my score.
> >
> > I like the table in appendix B where the author positioned their paper in the area. Please consider adding the new related works I mentioned  in the review. Maybe you can also update the related work section in the main paper.
> >
> > For BNNet notation, I am okay with your notations. Just please make sure they are consistent in your paper.

---

### Official Review · Reviewer_nqFj · 2023-07-26

**Soundness:** 3 good
**Presentation:** 2 fair
**Contribution:** 2 fair
**Rating:** 6
**Confidence:** 4

**Summary:**

The paper introduces a method for fine-tuning a client's model based on a pretrained model using its local data. The proposed personalization approach involves fine-tuning the learnable weights and biases (assume, there are $M$ such layers) as well as the means and standard deviations (SDs) of batch normalization (BN) layers (assume, there are $B$ BN layers).

This fine-tuning process is facilitated by two meta neural networks. The first meta neural network predicts $2M$ learning rates, which correspond to each weight and bias layer, enabling the local model to perform several gradient steps from the pretrained model. The input for this first meta neural network consists of the mean and standard deviation values for each of the $M$ layers.

On the other hand, the second meta neural network directly predicts the means and SDs. Its input comprises the distance between the pretrained means and the running means of the model with the pretrained weights on local data, as well as a similar distance measure between the SDs.

Both meta neural networks are trained using FedAvg on hyper gradients.

--------
Post-rebuttal update: I did not have significant concerns about this paper. I acknowledge the authors' rebuttal and extend my gratitude to them, particularly for adding the new experiment. I have opted not to modify my favorable rating of 6.

**Strengths:**

1. The paper demonstrates sufficient originality by introducing a novel approach using meta neural networks for computing personalized hyperparameters, as outlined in the summary above. Notably, the concept of trainable meta nets holds promise and appears to have practical applicability.

2. The authors effectively elaborate on the algorithm's intricacies, providing comprehensive descriptions of various components of the main method. Additionally, Figure 1 illustrates a schematic depiction of the main approach, enhancing the clarity of their explanations.

3. Extensive computational experiments are incorporated within the paper to substantiate the efficacy of the proposed method.

4. The paper is a valuable contribution to the fields of personalized Federated Learning and Meta Learning.

**Weaknesses:**

1. Despite the authors' efforts in providing extensive explanations for their ideas, regrettably, the text still contains several notable inconsistencies, as detailed in the "Questions" section.

2. A comparative analysis with the FOMAML method [1] is notably absent in the work. It is worth mentioning that FOMAML shares significant similarities with the authors' proposed method, as both utilize the MAML approach for fine-tuning the model using the training dataset and making optimization steps for the meta parameter based on the validation dataset, as observed in the current study. Additionally, in another related work [2], which builds upon MAML, the hyper gradient is explicitly addressed (see lines 14 and 16 of Algorithm 1 in [2]).

[1] "Model-Agnostic Meta-Learning for Fast Adaptation of Deep Networks", Finn et al.

[2] "Unsupervised Meta-Learning for Few-Shot Image Classification", Khodadadeh et al.

**Questions:**

1. Line 225: If I understand correctly, the text implies that $\partial_{{\omega}_{lr}} {\cal L}_V = 0$. If my interpretation is correct, could you please explain why this is the case?

2. Equation 3 & Line 199: What is the range of the index $i$?

Minor comments & suggestions:
1. Line 6: Depending -> depending
2. Line 35: Even -> even
4. Inaccuracy on lines 87-91: Although it is evident what the authors intended to convey, the wording is imprecise. I would recommend specifying the subset of chosen clients as $\tilde{C}$ on line 87, as done in Algorithm 1, and then using $i \in \tilde{C}$ instead of $i \in C$ on line 88. The statement $P_i = \cal{X} \times \cal{Y}$ on line 90 is also misleading: $\cal{X} \times \cal{Y}$ represents the domain of the distribution and does not define the distribution itself. Accordingly, the summation on line 91 should be taken over $i \in \tilde{C}$.
5. Line 93: The space before the word "for" is missing.
6. The first part of the statement on lines 93-95, in my opinion, is incorrect. From a statistical perspective, the aim of Federated Learning is to minimize the generalized local loss function $\mathbb{E}_{(x, y) \in P_i} \cal{L}_i (\theta, x, y)$. However, many Federated Learning frameworks minimize the finite sum because they make an implicit assumption that the distributions are similar. Otherwise, one could construct two distributions $P_1$ and $P_2$ such that the global solution for $1/2(P_1 + P_2)$ is ineffective for both clients.
7. The abbreviation SD is used on line 109 but only defined later on line 180.
8. Line 121: "are" -> "is"
10. Line 191: For clarity, I would suggest spelling out the abbreviation MLP.
11. Line 190: Please provide explicit definitions and domains for $\eta_i$ and $\beta_j$ as they are not scalars.
12. Equation 5 is misleading. The loss function on both lines is taken over the data distribution $P_i$, but there are also indices $T$ and $V$, which denote train and test datasets. $theta_i(\lambda)$ is not a minimizer of the generalized loss function but rather a fine-tuning of the global solution $\theta_g$. It takes some time to understand the authors' intended meaning here.
13. Line 226: There is a redundant full stop after the first word "Algorithm."
14. Algorithm 1, the input section: The phrases "Total no. of clients C. Num of update iterations K" use different notation to express "number," leading to inconsistency within the text.
15. Lines 9-10 in Algorithm 1 do not clearly depict the intended FedAvg iterates.
16. The document concludes on page 8 with Section "4.4 Personalizing to Different Domains."

**Limitations:**

Authors adequately address the limitations of the work.

---

> ### Author Rebuttal · Authors · 2023-08-07
>
> We thank the reviewer for their time and for providing constructive feedback to improve our work. We appreciate the questions and all the detailed minor comments and suggestions and will adjust them accordingly in the revised paper, fixing algorithmic, notation, and text ambiguities and errors.
>
> > Q1.1) Line 225: If I understand correctly, the text implies that $\partial\_{\boldsymbol{w}\_{lr}}\mathcal{L}\_{V} = 0$. If my interpretation is correct, could you please explain why this is the case?
>
> Yes, your understanding is correct. This is because validation loss does not directly depend on $\boldsymbol{w}\_{lr}$; there is no direct gradient as the learning rate used does not contribute to the validation loss. $\boldsymbol{w}\_{bn}$, on the other hand, is used in the forward pass in order to compute the validation loss, hence $\partial\_{\boldsymbol{w}\_{bn}} \mathcal{L}\_{V} \neq 0$.
>
> > Q1.2) Equation 3 & Line 199: What is the range of the index $i$?
>
> $i$ represents client $i \in C$. Hence, the range is the number of clients, $|C|$.
>
> > Q1.3) A comparative analysis with the FOMAML method [1] is notably absent in the work. It is worth mentioning that FOMAML shares significant similarities with the authors' proposed method, as both utilize the MAML approach for fine-tuning the model using the training dataset and making optimization steps for the meta parameter based on the validation dataset, as observed in the current study. Additionally, in another related work [2], which builds upon MAML, the hyper gradient is explicitly addressed (see lines 14 and 16 of Algorithm 1 in [2]).
>
> Thank you for the suggestion. The primary difference in MAML-based approaches and our approach is in the choice and design of the quantity to meta-optimize in the bilevel optimization – i.e., our meta-nets. MAML learns a **constant** initial condition of the inner loop of Eq. 5. We learn a **function** that predicts the hyperparameters of the Eq5. inner loop optimization from local client metadata (Eq. 4), e.g. the BN and learning rate hyperparameters. A secondary difference is our use of IFT to obtain meta-gradients, in contrast to FOMAML’s first order approximation to meta-gradient computation. We remark that the intersection of these points – our use of learning a learning rate prediction function with IFT – is noteworthy, because it is not trivial to learn learning rates directly with IFT meta-gradient as discussed in L243-246.
>
> We remark that off-the-shelf FOMAML (as an initial condition learner) is not a meaningful point of comparison because our problem is to fine-tune/personalize pre-trained models. Therefore, to compare with FOMAML, we focus on learning our FedL2P meta-nets with FOMAML style meta-gradients instead of IFT meta-gradients. In order to apply FOMAML to learning rate optimization, this also required extending FOMAML with the same trick as we did for our IFT approach as shown in Eq 9 & 10. However, FOMAML ignores the learning trajectory except the last step, which may result in performance degradation over longer horizons. We term this baseline FOMAML+.
>
> The table below, reports results on the multi-domain datasets as described in Section 4.4, keeping the same initial condition and meta representation (LRNet and BNNet meta-nets), and varying only the optimization algorithm.  Our approach of using IFT to compute the best response Jacobian, outperforms FOMAML+ for $\alpha=1000,1.0,0.5$ and has comparable performance for $\alpha=0.1$.
>
> | Dataset    | α    | +FT (BN C) | +FedL2P (FOMAML+) | +FedL2P (IFT) |
> |------------|------|------------|-------------------|---------------|
> | Caltech-10 | 1000 | 80.97±0.33 | 83.2±1.92         | 88.85±0.89    |
> | DomainNet  | 1000 | 52.17±1.55 | 52.7±0.17         | 54.38±0.45    |
> | DomainNet  | 1.0  | 62.27±0.58 | 63.14±0.38        | 63.77±0.44    |
> | DomainNet  | 0.5  | 71.39±0.97 | 71.37±0.79        | 72.64±0.3     |
> | DomainNet  | 0.1  | 86.03±0.47 | 86.22±0.16        | 86.36±0.45    |

---

> > ### Comment · Reviewer_nqFj · 2023-08-15
> > **Thanks for the reply**
> >
> > I extend my gratitude to the authors for their insightful rebuttal and for providing the accompanying table. Their comprehensive response is duly noted. I remain optimistic that the authors will fulfill their commitment to refining the manuscript in accordance with their assurances.

---

### Official Review · Reviewer_P9q9 · 2023-08-07

**Soundness:** 3 good
**Presentation:** 3 good
**Contribution:** 3 good
**Rating:** 5
**Confidence:** 3

**Summary:**

The paper proposes FedL2P, a federated learning framework for learning personalized strategies in the federated meta-learning problem. The authors introduce meta-nets to estimate hyper-parameters for personalized fine-tuning, such as batch normalization parameters and learning rates, based on client-specific data statistics. The empirical results show that FedL2P outperforms standard hand-crafted personalization baselines in both label and feature shift situations.

**Strengths:**

- The proposed method demonstrates soundness in its approach. Utilizing hyperparameter inference for federated learning shows promise and effectiveness in achieving personalization, as it tries to customize only a few crucial hyper-parameters.
- The experiments are meticulously evaluated across diverse distribution shifts, encompassing feature shift, label shift, and domain shift.

**Weaknesses:**

- Calculating meta-gradients involves time-consuming computations of second-order derivatives.
- The comparison is limited to variants of finetuning, neglecting the majority of personalized federated learning baselines such as [1,2]. Additionally, recent works [3] addressing distribution shifts by optimizing different blocks of the network should be thoroughly discussed in the paper.
- The sufficiency and effectiveness of communicating the selected hyper-parameters for personalized federated learning lack rigorous verification.

[1] Towards personalized federated learning. TNNLS

[2] Layer-wised model aggregation for personalized federated learning. CVPR 2022

[3] Surgical fine-tuning improves adaptation to distribution shifts. ICLR 2023

**Questions:**

- What is the computational cost associated with the proposed method?
- How many parameters will be personalized in this study?
- Could you provide insights into the learning rate distribution for different blocks/layers in the paper and elaborate on what the model has learned? This would greatly enhance the understanding of the results.


**Limitations:**

I'm not aware of any potential negative societal impact of the work.

---

> ### Author Rebuttal · Authors · 2023-08-07
>
> Thank you for the detailed comments and feedback. We hope our response below will address the reviewer’s concerns and questions. We will revise the paper acoordingly to include these discussions.
>
> > Q5.1) Calculating meta-gradients involves time-consuming computations of second-order derivatives. What is the computational cost associated with the proposed method?
>
> The computation cost of approximating the Hessian and computing the hypergradient is detailed in Appendix C. To reiterate, our proposed method takes 0.24 seconds to compute the hypergradient as compared to FOMAML, which takes 0.08 seconds, and HF-MAML, which takes 0.16 seconds, to compute the meta-gradient. Hence, our approach does not incur a significant overhead compared to these simpler non-Hessian methods. During fine-tuning, our proposed method incurs a minor additional cost of 4.4% more than regular fine-tuning.
>
> > Q5.2) The comparison is limited to variants of finetuning, neglecting the majority of personalized federated learning baselines such as [1,2]. Additionally, recent works [3] addressing distribution shifts by optimizing different blocks of the network should be thoroughly discussed in the paper.
>
> [1] surveys a wide range of methods used for personalized FL. The majority of the works listed in this paper can be mostly improved via further local fine-tuning as shown in recent large-scale personalized FL benchmark papers [9,43 in the paper]. As our proposed method learns to improve the fine-tuning process, it is complementary to many of these works. In Table 1, we showed three representative cases, namely FedAvg, PerFedAvg, and FedBABU. FedAvg optimizes on the initial global accuracy, PerFedAvg optimizes directly on the personalized accuracy after fine-tuning, and FedBABU improves both initial and personalized accuracy through model splitting (global encoder + local prediction heads). Our FedL2P is shown to improve on all the above.
>
> In Appendix B, we discussed the scalability of a wide range of existing personalized FL works and Appendix C details the cost of our approach in comparison with existing works that require storing of local models and hypernetworks to generate per-client models. To reiterate, in relation to the reviewer’s suggested baseline [2], hypernetwork-based approaches scale poorly with the number of clients and the model size. In our experiments with pFedHN, the hypernetwork fails to generate reasonable client weights as it does not scale to larger ResNets used in our experiments. Note that pFedHN and [2] used a toy model of 5 layers with a maximum of 100 clients whereas we used a full-blown ResNet with a maximum of 1000 clients.
>
> [3], along with other papers that fine-tunes a different subset of the model [46,11,14,36,16], manually handcraft a selection of layers to freeze and personalize. Specifically, [3] showed that, depending on the dataset and the pretrained model, freezing and updating different parts of the model showed performance gains. In contrast, our approach, automatically learns not only which layers to freeze/update but also the learning rate of each layer according to the specific scenario for each specific client, namely the client’s data distribution and dataset in relation to the pretrained model. Hence, our approach encompasses, generalizes, and automates many previous manually designed heuristics for layer-wise updates.
>
> > Q5.3) The sufficiency and effectiveness of communicating the selected hyper-parameters for personalized federated learning lack rigorous verification. Could you provide insights into the learning rate distribution for different blocks/layers in the paper and elaborate on what the model has learned? This would greatly enhance the understanding of the results. How many parameters will be personalized in this study?
>
> In our results, the learning rates of different layers is heavily dependent on the scenario and client. While previous handcrafted solutions mentioned in Q5.2) surgically freeze specific parts (e.g. first block/middle block/last layer) of the model for **all** clients, we observed that, in most cases, our approach often freezes and updates layers of various different parts of the model with different layer-wise learning rates for **each** client. For instance, in our CIFAR-10 ($\alpha=0.1$) setup, our approach learns highly sparse models (~0.9 as shown in Figure 2a), updating the first convolution layer of the 1st, 2nd, and 3rd block, along with the last layer weights and 2nd convolution layer of the 2nd block depending on the client. An exception to this is found in our CIFAR-10 IID ($\alpha=1000$) setup, where our approach learns to freeze the global model (0 parameters personalized) as seen in Figure 2a ($\eta=0$ and model sparsity = 1). Similarly, in our Office-Caltech-10 and DomainNet experiments, our approach learns to update all layers of the model to adapt to the feature and concept distribution drift as seen in Figure 2c & d (model sparsity = 0). All in all, the number of parameters, which parameters, and the extent in which we update the parameters depends on the scenario and the client.

---

> > ### Comment · Reviewer_P9q9 · 2023-08-20
> > **Response to Rebuttal**
> >
> > Thanks for your detailed rebuttal. I'll keep my rating.

---

### Author Rebuttal · Authors · 2023-08-08

We would like to express our appreciation and gratitude to the reviewers for the detailed comments and helpful feedback to further clarify and improve our paper. We will revise the paper accordingly, adding the new evaluations, clarifying ambiguities, and fixing notation & spelling errors. We have attached a Rebuttal PDF, which consists of results on our newly added experiments on the Speech Commands dataset (in response to Reviewer wFN9 Q3.4 and Reviewer Jjav Q4.4.) and an extended Table 1 to highlight the improvement gains of FedL2P compared to standard fine-tuning in response to Reviewer wFN9 Q3.5.

---

### Decision · Program_Chairs · 2023-09-21

**Decision:**

Accept (poster)

**Comment:**

All reviewers found the work interesting and relevant. After the author’s response, the reviewers did not see the need for further clarifications. All reviewers recommend acceptance.